Journal of Data-centric Machine Learning Research (2025) Submitted 1/11; Revised 3/30; Revised 9/07; Published 9/17

# Towards Causal Relationship in Indefinite Data:
# New Datasets and Baseline Model

**Hang Chen**                                               ALBERT2123@STU.XJTU.EDU.CN
*Xi'an Jiaotong University*

**Xinyu Yang**[*]                                           YXYPHD@MAIL.XJTU.EDU.CN
*Xi'an Jiaotong University*

**Keqing Du**                                               DUKEQING@STU.XJTU.EDU.CN
*Xi'an Jiaotong University*

**Reviewed on OpenReview:** *https://openreview.net/forum?id=LxROMRFBbS*

**Editor:** Mykola Pechenizkiy

## Abstract

The cross-fertilization of deep learning and causal discovery has given birth to broader causal data forms, involving **multi-structured data** like the Netsim dataset, and **complex variables** such as those in the RECCON dataset. Interestingly, we observe an absence of research that concurrently addresses data with multi-structures and complex variables, named 'indefinite data.' In our previous survey, we introduced the concept of this data paradigm, yet the exploration of indefinite data still faces two substantial gaps: the dataset gap and the model gap. In this paper, we release two high-quality datasets - *Causalogue* and *Causaction* for dataset gap, containing text dialogue samples and video action samples with causal annotations respectively. Moreover, the model gap arises from the coexistence of multi-structure data and complex variables, breaking the assumptions of all current methods, and rendering them infeasible on indefinite data. To this end, we propose a probabilistic framework as a baseline. It enables overcoming challenges brought by indefinite data, and paves the way for the extension of latent confounders. Comprehensive experiments have evaluated baseline results of causal structures, causal representations, and confounding disentanglement. Our codes and datasets are available at Github URL.

**Keywords:** Causal Dataset, Causal Representation, Causal Structures, Baseline Model

## 1 Introduction

In light of the recent advances in deep learning, there is a growing tendency to incorporate causal discovery in more complex forms of data, including images (Jerzak et al., 2022; Ribeiro et al., 2023), text (Zhang et al., 2023), and videos (Bagi et al., 2023). Generally, there are two purposes for these incorporations: one is to uncover the underlying **causal structure** (Sun et al., 2023; Li et al., 2023; Golan and Foley, 2023; Squires et al., 2022;

---

*. Corresponding author: yxyphd@mail.xjtu.edu.cn

Zhang et al., 2017) within the data, the other is to learn effective **causal representations** (Wang et al., 2023; Olesen, 1993; Xu et al., 2023; Li and Fu, 2014; Liu et al., 2023; Balashankar and Subramanian, 2021).

Our recent work (Chen et al., 2023a) has summarized different data forms of these incorporations based on causal structure and causal representation respectively. Regarding causal structure, there are **single-structure** data (Pearl et al., 2000; Yu et al., 2019; Lachapelle et al., 2019) and **multi-structure** data (Lorch et al., 2022a; Ke et al., 2020; Löwe et al., 2022), depending on whether multiple causal structures (causal graphs) are involved in the dataset or task. For example, fMRI dataset (Smith et al., 2011) suggests the different brain region activity levels of Patient $A$ and $B$ as two samples, corresponding to two causal structures. Concerning causal representation, there are **simple variables** representations (Zhou et al., 2022; Cai et al., 2019; Tank et al., 2022) and **complex variables** representations (Li et al., 2021; Zhang et al., 2022; Oh et al., 2021), depending on whether the causal variables inherently have a numerical form. Variables like age, height, weight, blood pressure is typically treated as simple variables (Guvenir et al., 1997), while a sentence (Chen et al., 2023b) or a video (Du et al., 2023) are non-numerical and unstructured, as well as they often needs to be converted by deep models into high-dimensional continuous representation (such as sentence embeddings or optical flows) to make them calculable, thus named as complex variables.

These data forms highlight the intrinsic differences in the applied algorithms. Algorithms applied to multi-structure data consider factors such as amortized causal discovery (Löwe et al., 2022) and sample utilization rate, whereas algorithms applied to complex variables explore how to overcome obstacles presented by the unknown distribution of high-dimensional continuous representations.

Moreover, our work (Chen et al., 2023a) conjectured the emergence of a new causal data paradigm - **indefinite data** with the characteristics of both **multi-structure data** and **complex variables**. For instance, taking a piece of dialogue (including $N$ utterances as $N$ causal variables) as an input, could we recover the complete causal relationships between these utterances and learn each utterance's causal representation for causality-related downstream tasks (like ECPE task (Xia and Ding, 2019))? Or, if we replace the dialogue with the circuit graph (including $N$ gates as $N$ causal variables), could we learn the internal relationships among these gates to reflect who contributes most to potential changes and their corresponding causal representations for learning functions?

Despite the comprehensive definition provided by our previous survey (Chen et al., 2023a), the study of indefinite data still faces two research gaps: the **dataset gap** and the **model gap**. Specifically, causal relationships in indefinite data are often obscure and subjective, making it challenging to collect samples with obvious causal relationships and objective annotations. Moreover, the co-occurrence of multi-structure data and complex variables breaks the hypotheses of all existing methods' frameworks, necessitating a redesign of how causal representations and causal structures can be simultaneously learned.

To overcome these research gaps, we aim to release two high-quality indefinite datasets and a baseline model, specifically:

In **Section 3**, we investigated the existing datasets and introduced two brand-new indefinite datasets - *Causalogue* and *Causaction*. *Causalogue* is a **text** dataset containing dialogue samples used for analyzing causal relationships between utterances. To ensure the

causal relationships are apparent and objective, we utilized GPT-4 to generate dialogues according to pre-defined causal patterns. *Causaction* is a **video** dataset containing different action segments, used for analyzing the causal relationships between different actions within a video. Annotators were asked to judge causal relationships directly based on low-level labels, rather than judging each video sample, thereby significantly enhancing the consistency of causal relationships.

In **Section 4**, we introduced the existing model framework on structure learning and representation learning to explain why these models does not enable indefinite data well. Hence, we proposed a probabilistic framework as a baseline model for indefinite data based on Structural Causal Models (SCMs). It combines and then simplifies the frameworks of structural learning and representation learning. Although there may be a risk of causal inconsistency (we discuss this issue in the Discussion, i.e., Section 6), it can simultaneously learn the causal mechanisms of multi-structure and complex variables. Moreover, the estimation of confounding effects disentangle the causal representation and the confounding representation, enabling the model to adapt to data with latent confounders.

In **Section 5**, we designed comprehensive evaluation metrics for indefinite data on causal representation and causal structures, and compared them with some of the most adaptable methods. Additionally, to directly evaluate the performance of deconfounding, we also created a synthetic dataset with a known confounding distribution.

To the best of our knowledge, this paper contributes two 'firsts':

- The introduction of the first 'Indefinite' datasets—*Causalogue* and *Causaction*—with comprehensive causal labels, providing a data foundation for the research of indefinite data.

- The proposal of the first variational model framework adaptive to indefinite data, complemented by baseline metrics on the above 2 datasets.

## 2 Preliminaries

### 2.1 Input-Output Framework

Suppose $X_i$ is any sample in the dataset, which contains $N$ causal variables, i.e., $X_i = \{x_{i,1}, ..., x_{i,N}\}$. Through a specific causal model, it generates two outputs: the causal structure $\hat{\mathcal{G}}_i$ and the causal representation $\hat{X}_i$[1]. Here, $\hat{\mathcal{G}}_i$ represents the causal relationships between $N$ variables, existing in the form of a DAG, also known as a causal graph. $\hat{X}_i \in R^{N*d}$ represents $N$ $d$-dimensional causal representations of $X_i$, entailing the information of underlying and abstract causal relations from perceptible input, which can be subsequently used for classification, prediction, decision, etc. We formalize such a causal discovery model as:

$$\hat{X}_i, \hat{\mathcal{G}}_i = CausalModel(X_i) \tag{1}$$

---

1. We thank the reviewers for their valuable feedback. It is important to note that the "causal representation" referred to in this paper differs from that in some other studies (Schölkopf et al., 2021). Here, "causal representation" can be understood as a representation of the input $X_i$ that encodes causal information, sharing the same domain as the work in (Yu et al., 2019; Löwe et al., 2022; Lorch et al., 2022a).

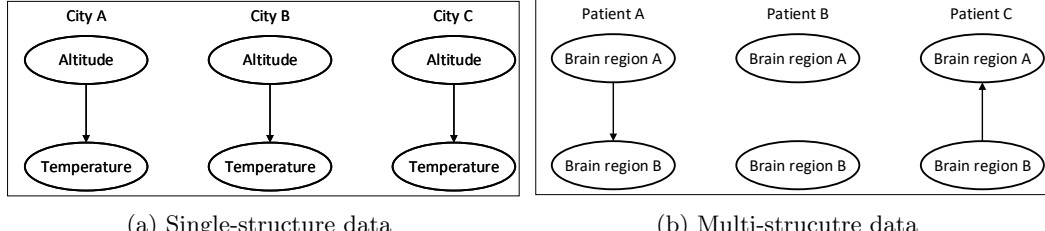

(a) Single-structure data    (b) Multi-strucutre data

Figure 1: Comparison between single-structure data and multi-structure data.

## 2.2 Categories in Structures and Representations

Following the settings in (Chen et al., 2023a), we have classified the data. Structurally, there are single-structure data and multi-structure data (Löwe et al., 2022). In terms of representation, there are simple variables and complex variables. We provide specific descriptions of these four types as follows:

**Single-structure data**: All samples in a dataset share a single causal structure. That is, for any sample $X_i$, the correct model should estimate a completely identical $\hat{\mathcal{G}}$.

**Multi-structure data**: There are $M$ causal structures ($M > 1$) in the entire dataset. That is, for any sample $X_i$, the correct model should estimate one of the $M$ structures $\hat{\mathcal{G}}_m \in \{\hat{\mathcal{G}}_1, ... \hat{\mathcal{G}}_M\}$.

**Simple variables**: Variables that exist in samples in a fixed numerical form, such as blood pressure, temperature, age, etc. Most of the time, the values of such variables themselves are used as causal representations without the need for additional calculations, i.e., in the input-output framework, $\hat{X} = X$.

**Complex variables**: Variables that do not have a fixed numerical form in samples, such as text, video, and data in other modalities. The input $X$ is some initial representation of it (such as embeddings), and the correct model should apply a causal mechanism to it, therefore usually $\hat{X} \neq X^2$.

It's worth noting that multi-structure data does not imply that each sample corresponds to multiple causal structures. In Figure 1, we list three samples of single-structure data and multi-structure data respectively. In single-structure data, we observe the influence of the variable "Altitude" on the variable "Temperature". As a physical law, it does not change with the sample. However, in multi-structure data, we observed the influence of the variable "Brain region A" on the variable "Brain region B". Since different patient samples have different symptoms, these symptoms cause different electrical signal response rules between brain areas, i.e., different causal structures.

Additionally, we have also compared simple variables and complex variables in Table 1. For simple variables, we select a sample containing age and a sample containing voltage. For complex variables, we select a sample containing text and a sample containing video. Obviously, for simple variables, $\hat{X} = X$, so effective verification for can be carried out for causal representation. However, for complex variables, $\hat{X} \neq X$, because the tensor of $X$ is an initial representation representing the numerical value of the variable while the

---

2. In the scenario of simple variables, with small probability, it is necessary to calculate a representation for such variables that is not equal to the original value. In this case, they are treated as complex variables.

Table 1: Comparison between simple variables and complex variables

| Category | Variables | $X$ | $\hat{X}$ | Dimension(d) |
|---|---|---|---|---|
| simple variable | Age | 25 | 25 | $d = 1$ |
| | Voltage | 2 | 2 | $d = 1$ |
| complex variable | Token | tensor | tensor | $d > 1$ |
| | Frame | tensor | tensor | $d > 1$ |

tensor of $\hat{X}$ is causal representation including complex causal relationships between context variables. Due to the high-dimensional continuous representation, it is difficult to verify causal representation through these statistics-based methods.

Therefore, in this paper, we default to using neural networks transferring representations to relationships to validate the causal representation of complex variables. Specifically, for any two variables in any sample $X_s$ arranged in order $< x_{s,i}, x_{s,j} >$, if there exists a causal relationship $x_{s,i} \to x_{s,j}$, we have label $Y_{<x_{s,i},x_{s,j}>} = 1$, otherwise, we set $Y_{<x_{s,i},x_{s,j}>} = 0$. Let $f_c$ be a causal classifier, e.g., an MLP followed by a sigmoid function, with the input being the causal representation of any two variables $< \hat{x}_{s,i}, \hat{x}_{s,j} >$, and it is supervised and trained according to the label $Y_{<x_{s,i},x_{s,j}>}$. During validation, if $f_c(< \hat{x}_{s,i}, \hat{x}_{s,j} >) \in (0, 0.5]$, it indicates there is no relationship pointing from $x_{s,i}$ to $x_{s,j}$, and if $f_c(< \hat{x}_{s,i}, \hat{x}_{s,j} >) \in (0.5, 1)$, there is a relationship $x_{s,i} \to x_{s,j}$. The experimental validation of $f_c$ is provided in Appendix A.

From above basic types of causal data, the new data paradigm named "indefinite data" (Chen et al., 2023a) is defined as:

**Definition 1** (indefinite data). *The causal relationships exist in a dataset $\mathbf{D} = \{X_s\}_{s=1}^S$ which has $S$ samples and $M$ ($M > 1$) causal structures ($\mathcal{G} = \{\mathcal{G}_m\}_{m=1}^M$). Each structure is formalized as a graph, i.e., $\mathcal{G}_m = (X_m, \mathcal{E}_m)$ (where $X_m$ stands for the nodes and $\mathcal{E}_m$ represents edges). Hence, each sample $X_{s,m} \in \mathbb{R}^{N_m \times d}$ ($d > 1$) belongs to a causal structure $\mathcal{G}_m$ and consists of $N_m$ variables: $X_{s,m} = \{x_{s,m,n}\}_{n_m=1}^{N_m}$. $\hat{X}_{s,m} \in \mathbb{R}^{N_m \times d}$ represents the causal representation of $X_{s,m}$.*

Generally, the identifiability of causal relationships is guaranteed by acyclic constraints (e.g., NOTEARS (Zheng et al., 2018)) and independent noise (introduced by SCM). In this work, we assume that all causal models are extensions of the fundamental Structural Causal Model (SCM). Consequently, identifiability is solely dependent on acyclicity constraints. For simplification, we assume that all indefinite data comply with the time order (indeed, in this paper, all the indefinite datasets adhere to this assumption, and we compare the performance with acyclic constraints in Appendix B).

**Assumption 2** (Causal Identifiability). *The index of causal variables in indefinite data sample $X_{s,m}$ satisfies time order, defined as a linear order $\prec_{X_{s,m}}$. Let $\mathcal{J}_{X_{s,m}}$ be the index set of $X_{s,m}$. $\forall i, j \in \mathcal{J}_{X_{s,m}}$, if $i < j$, $x_i \prec_{X_{s,m}} x_j$.*

Therefore, the causal order can be regarded as a partial order, defined as $\preccurlyeq_{X_{s,m}}$ w.r.t. the time order $\prec_{X_{s,m}}$. That is, $\forall x_i \prec_{X_{s,m}} x_j$, there must be $x_i \preccurlyeq_{X_{s,m}} x_j$.

For instance, according to the input-output framework, if the input is a dialogue containing utterances, as shown in Figure 2, and the task requires estimating the causal relationships of these 4 utterances and their corresponding causal representation (Poria et al.,

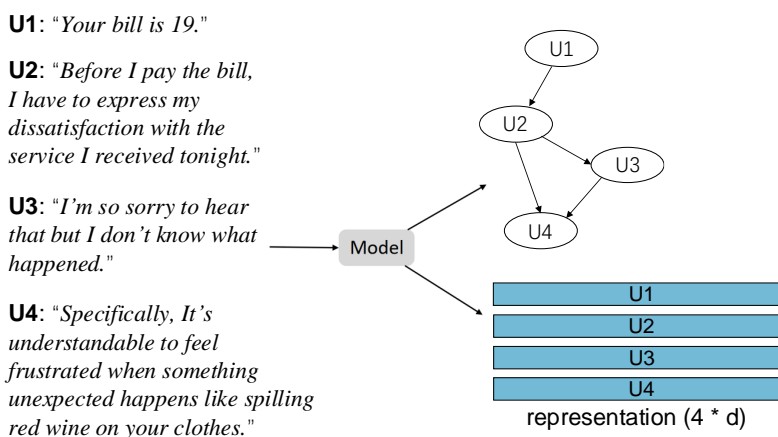

**U1**: *"Your bill is 19."*

**U2**: *"Before I pay the bill, I have to express my dissatisfaction with the service I received tonight."*

**U3**: *"I'm so sorry to hear that but I don't know what happened."*

**U4**: *"Specifically, It's understandable to feel frustrated when something unexpected happens like spilling red wine on your clothes."*

representation (4 * d)

Figure 2: Sample from Causalogue dataset, applied in our input-output framework.

Table 2: Comparisons between causal datasets

| Datasets | Multi-structure | Complex Variables | DAG | Branch | Multi-hop |
|---|---|---|---|---|---|
| Net-fmri | ✓ | ✗ | ✓ | ✓ | ✓ |
| C-MINIST | ✗ | ✓ | ✓ | ✓ | ✗ |
| GSM8K | ✓ | ✓ | ✗ | ✗ | ✗ |
| SVAMP | ✓ | ✓ | ✗ | ✗ | ✗ |
| Corr2Cause | ✓ | ✓ | ✓ | ✗ | ✗ |
| CLadder | ✓ | ✓ | ✓ | ✗ | ✗ |
| CausalDialogue | ✓ | ✓ | ✓ | ✓ | ✗ |
| *Causalogue* | ✓ | ✓ | ✓ | ✓ | ✓ |
| *Causaction* | ✓ | ✓ | ✓ | ✓ | ✓ |

2021), then $\hat{\mathcal{G}}$ is equivalent to a directed acyclic graph (DAG) containing 4 nodes, representing the causal relations between 4 utterances, and $\hat{X} \in R^{4*d}$, where $d$ is the dimension of the representation, can represent the causal representation of each utterance. The causal representation is validated through $f_c$.

## 3 Indefinite Datasets

### 3.1 Comparison with Related Work

In this section, we have selected representative datasets from various types of causal datasets, including: Multi-structure: Net-fmri (Smith et al., 2011), Complex Variables: C-MINIST (Fan et al., 2022), Mathematics and Reasoning: GSM8K (Cobbe et al., 2021) and SVAMP (Patel et al., 2021), Large Model Reasoning: Corr2Cause (Jin et al., 2024), CLadder (Jin et al., 2023), Dialogue: CausalDialogue (Tuan et al., 2023), and our datasets: *Causalogue* and *Causaction*. For these datasets, we investigated five aspects: whether they are multi-structure datasets (Multi-structure), whether the causal variables are complex variables (Complex Variables), whether they have complete causal graph labels (DAG), whether the causal graph has branches (Branch), and whether the causal graph contains edges between multi-hop nodes (Multi-hop). Among these aspects, Multi-structure and Complex Variables measure the complexity of the data, while DAG, Branch, and Multi-hop measure the completeness of the implied causal structures.

Table 2 presents the evaluation of these datasets in five aspects. An interesting finding is that complete causal structure annotations (DAG, Branch, Multi-hop) tend to appear in simple data types, and it is more difficult to find complete causal structure annotations in complex data types (such as indefinite data). Obviously, on these datasets with indefinite data, most causal relationships are obscure, which leads to poor consistency in manual annotation. Taking dialogue as an example, utterances that have not been observed before might likely act as confounding factors influencing the correlation between observed utterances. Moreover, the standards for judging whether there is a causal relationship between two utterances is terribly subjective. Up to now, only a fraction of the work (Poria et al., 2021; Chen et al., 2023c) has annotated some evident causal relationships, and no complete causal-labeled dataset has yet to appear, which significantly dampens researchers' enthusiasm for indefinite data.

### 3.2 *Causalogue*

#### 3.2.1 Attributes

*Causalogue*[3] is the first dialogue dataset that includes comprehensive causal relationship labels for indefinite data. Additionally, we employ GPT-4 generation as a substitute for data collection from the real world or manual simulation, which considerably mitigates the presence of obscure causal structures. In Appendix C, we provide an analysis of the challenges associated with achieving consistent annotations for both LLM-generated and wild text, as well as the increased difficulty introduced by varying the number of causal variables. This analysis motivates our choice to focus on LLM-generated samples with four causal variables as the target scale for our dataset.

The dataset incorporates 10 types of causal structures (M = 10), each with several samples (Detailed numbers are presented in Table 3, "Small" signifies samples that have been manually checked, while "large" refers to all samples generated by GPT-4 without manual verification). The detailed attributes are following:

**Causal Variables**: We treat each dialogue as a sample, comprised of 4 utterances, which we define as 4 causal variables. Further, the first and third utterances originate from the same speaker, defined as $speaker1$. Similarly, the second and fourth utterances are from another individual, defined as $speaker2$.

**Causal Relationship**: In each sample, binary causal relationships have been labeled between any two utterances-"1"represents that there exists a causal relationship and "0" represents there is not.

**Structure**: We have designed 10 types of causal structures in the dataset as shown in Figure 3. Taking the Chain_II as an example, this model adds an additional causal relationship from $Utt_1 \rightarrow Utt_3$ based on the Chain_I, indicating that $Utt_3$ considers not just the effect from $Utt_2$ but also from $Utt_1$.

**Sample**: We consider a dialogue as a sample, with each sample comprising 4 utterances representing 4 causal variables. Each sample corresponds to one of the 10 causal structures outlined above, annotating whether a causal relationship exists between any two utterances.

---

3. We have included a separate PDF in the supplementary material to supplement the datasheets describing the new dataset.

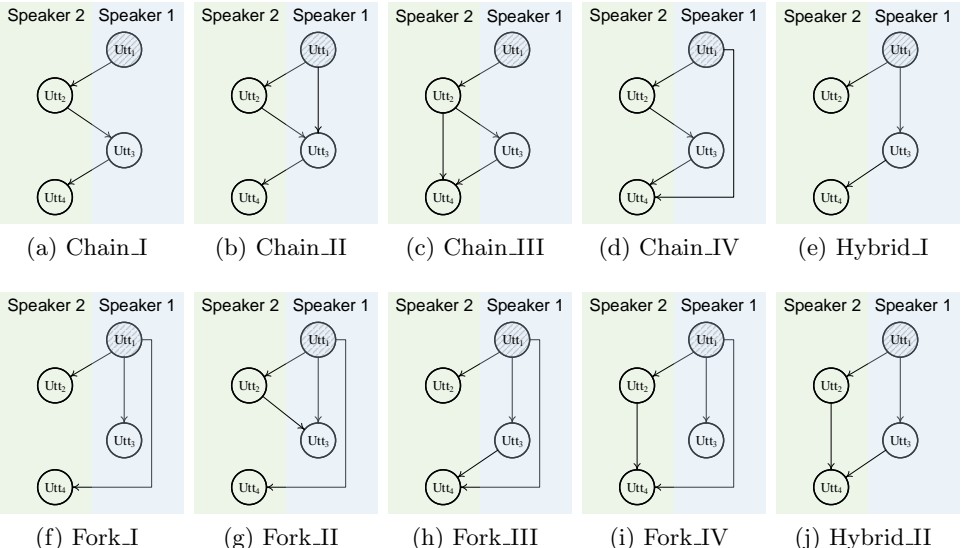

Figure 3: 10 structures in *Causalogue* Dataset.

Table 3: Number of the samples in *Causalogue* Dataset

| Versions | Structure Types | | | | | | | | | | |
|---|---|---|---|---|---|---|---|---|---|---|---|
| | Chain_I | Chain_II | Chain_III | Chain_IV | Fork_I | Fork_II | Fork_III | Fork_IV | Hybrid_I | Hybrid_II | Total |
| Small | 276 | 84 | 141 | 44 | 257 | 237 | 251 | 67 | 185 | 77 | 1638 |
| Large | 0 | 524 | 508 | 513 | 1215 | 645 | 501 | 372 | 499 | 635 | 5412 |

Due to Assumption 2, our labels only consider forward-causal relationships. An example of a Chain_III sample is shown as follows:

"causal_type": "Chain_III",

"clause": { "1": "Your bill is 19.", "2": "Before I pay the bill, I have to express my dissatisfaction with the service I received tonight.", "3": "I'm so sorry to hear that but I don't know what happened.", "4": "Specifically, It's understandable to feel frustrated when something unexpected happens like spilling red wine on your clothes."},

"dia_id": 1,

"label": { "1": "0,0,0,0", "2": "1,0,0,0", "3": "0,1,0,0", "4": "0,1,1,0"}

In the given example, the $Utt_4$ serves as a response to the $Utt_3$, while simultaneously attaching to the speaker's $Utt_2$—thereby rendering both the $Utt_2$ and $Utt_3$ as causes to the $Utt_4$. Indeed, during the generation process of the $Utt_4$, we made sure to inform GPT-4 of the existence of $Utt_2$ and $Utt_3$.

### 3.2.2 Creation Process

We utilized the API interface of GPT-4 [4] to defined the following variables: "*role*", which has three types - "*system*", "*user*", and "*assistant*". Here, "*system*" represents the background

---

4. https://platform.openai.com/docs/models/gpt-4

or a prior settings, while "*user*" and "*assistant*" are defined as speakers with two different identities. Additionally, the first utterance is pre-set. Hence, creating a dialogue requires a given combination: a fixed *first_utterance*, a specified *system* information, and a setting in which previous utterances are considered. We have a total of 149 *first_utterance* options, and there are as many as 278,867 combinations of *first_utterance* and *system* settings (our final samples only number in the 1638, to preserve the diversity and distinctiveness of our dialogues). What follows is an example of generating the third utterance in the structure of ChainII:

{*"role": "system", "content": "You are Peter, you have promised to go to a Chinese Opera with your daughter, so you want to have dinner with your friends in next Sunday."*}

{*"role": "assistant", "content": "Yes. Sunday sounds fine. What time?" (pre-set Utt_1)*}

{*"role": "user", "content": Utt_2*}

Upon creation, the samples are initially auto-annotated based on their designed labels, and then manually verified to ensure their validity. Our manual verification employed two annotators, who demonstrated proficient English understanding and communication skills, possessing sufficient knowledge about causality. The annotation consistency between these two annotators was tested through 833 samples, achieving a kappa coefficient of 0.92.

During the annotation process, if a sample was labeled differently by the two annotators, that sample was considered to possess an ambiguous causal relationship and thus was excluded from the final dataset. Only samples that were consistently labeled by both annotators were ultimately accepted.

Furthermore, to guarantee the freedom of manual annotation, we allowed the annotators to label structures that fell outside the predefined 10 causal structures. Specifically, we only requested annotators to judge whether any two utterances (satisfying Assumption 2) have a causal relationship, allowing them some discretion, which inevitably produced samples not belonging to the 10 causal structures. We classified these as the "Other" category.

The accuracy of labels was significantly improved after the manual annotation process. However, considering that the unverified samples might be utilized for other research areas, such as the ability of LLMs to focus on context, we have released two versions of the datasets, as demonstrated in Table 3. "Small" signifies samples that have been manually checked as correctly labeled, while "large" refers to all samples generated by GPT-4 without manual verification. We do not recommend considering the "large" version when undertaking causality-related work. Likewise, we have not taken it into our experiments.

### 3.3 *Causaction*

#### 3.3.1 Attributes

*Causaction* is another indefinite dataset that we obtained after re-annotating the Breakfast Dataset (Kuehne et al., 2014)[5]. It contains a total of 1,118 videos, documenting 10 different

---

5. Compared to other alternative datasets, such as the MPII Cooking dataset (Rohrbach et al., 2012) and 50salads (Stein and McKenna, 2013), we found in our annotation tests that the samples from the breakfast dataset have higher consistency. This may be because the causal relationships between actions in the breakfast dataset are less likely to cause controversy.

breakfast preparation processes (such as coffee, salad, sandwich, etc.). Each video consists of 4-9 actions, with a clear frame boundary. We have annotated the causal relationship between any two actions in a sample. Specific attributes are as follows:

**Causal Variable**: We treat each video as a sample, comprised of 4-9 actions as the causal variables. For simplicity, we follow the setting of MS-TCN (Farha and Gall, 2019), replacing the video resource of each action with pre-trained representation of I3D (Carreira and Zisserman, 2017).

**Causal Relationship**: According to the Assumption 2, we deem the time order of these actions in certain videos as a natural linear order. Hence, binary causal relationships have been labeled between any two actions satisfying the linear order ('0' represents there is no causal relationship while '1' represents there is). For example, process "cereals" includes 4 actions: "take bowl", "pour cereals", "pour milk", and "stir cereals". The all causal relationships labeled with "1" are: "take bowl → pour cereals", "take bowl → pour milk", "take bowl → stir cereals", "pour cereals → stir cereals", and "pour milk → stir cereals".

**Structure**: Unlike *Causalogue*, although the entire dataset includes the 10 types of preparation processes of breakfasts, the number of causal structures far exceeds 10. Most videos do not encompass all the actions in a process. For example, the entire process of "Salad" consists of 7 actions, but some videos are missing the "take plate" action, and some videos include the actions "cut fruit1" and "cut fruit 2".

**Sample**: We consider a video as a sample. The statistics of samples with different processes are shown in Table 4.

### 3.3.2 Creation Process

The original Breakfast Dataset has annotated the frame boundaries of each action. Therefore, in our annotation work, we don't need to ascertain which frames a causal variable contains. The annotators were asked to directly annotate at the action level to avoid inconsistencies caused by the subjectivity of watching videos. For example, in the coffee process, there are 6 actions, so a total of 15 binary relationship pairs need to be annotated. Specifically, we informed the annotators of the time order and the explanation of all actions in each process. After ensuring the understanding of each action, the annotators conducted a causal relationship evaluation on binary action pairs $(A, B)$ that satisfy the linear order relation, where 1 signifies a belief that action $A$ has a causal relationship with action $B$, and 0 represents no such relationship. An action pair is considered to have a causal relationship if the following conditions are met:

Table 4: The number of samples in *Causaction* Dataset

| Process | Number of Actions (Variables) | | | | | | |
|---|---|---|---|---|---|---|---|
| | 4 | 5 | 6 | 7 | 8 | 9 | all |
| cereals | 36 | - | - | - | - | - | 36 |
| coffee | 12 | 28 | - | - | - | - | 40 |
| friedegg | 52 | 45 | 53 | 7 | 4 | - | 161 |
| milk | 56 | 14 | 4 | - | - | - | 74 |
| salad | 6 | 52 | 29 | 25 | 37 | 33 | 182 |
| sandwich | 52 | 11 | 6 | 2 | 4 | - | 75 |
| tea | 14 | 5 | - | - | - | - | 19 |
| pancake | - | 100 | 26 | 24 | 33 | 41 | 224 |
| scrambledegg | 8 | 36 | 30 | 42 | 33 | 24 | 173 |
| juice | 65 | 36 | 24 | 6 | - | 3 | 134 |
| all | 301 | 327 | 172 | 106 | 111 | 101 | 1118 |

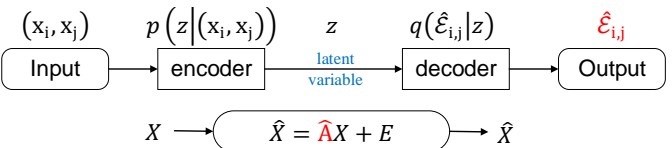

Figure 4: Generative framework to solve multi-structure and simple-variable data.

- According to life experience, after action $A$ happens, action $B$ is high-probably to occur.

- According to life experience, after action $B$ happens, action A is low probability to occur.

Finally, we binarize all annotation results, that is, binary pairs with a mean $> 0.5$ are marked as 1, and those with a mean $< 0.5$ are marked as 0.

The annotators consist of 10 researchers in the field of causal inference (Group A) and 217 deep-learning researchers (Group B). Initially, we asked Group A to annotate the two simplest processes, "milk" and "coffee," and considered their annotation results as the gold standard. Members of Group B first annotated "milk" and "coffee," with only those members having $> 80\%$ consistency with Group A deemed qualified. In the end, 190 qualified members were confirmed in Group B, joining the 10 members in Group A to form Group C (total of 200 members). Group C annotated the remaining 8 processes, and the statistical results after binarization were used as the final labels. During the annotation process, the consistency was 94.13% for Group A, 88.46% for the qualified members of Group B, and 88.74% for Group C.

## 4 Baseline Model

### 4.1 Related Work

indefinite data (Multiple structures & complex variables) can be simply considered as an integration of two types of "Multiple structures & simple variables" and "Single structure & complex variables". We introduce the related works about these two types of causal data as follows.

#### 4.1.1 Causal Structure Learning

In causal structure learning, causal variables are typically simple variables, i.e., in our input-output framework, the output $\hat{X}$ should equal $X$. Therefore, for single-structure data, a large body of work has demonstrated that accurate statistics between variables can be used to recover the causal structure (Kalisch and Bühlman, 2007; Chickering, 2002; Hauser and Bühlmann, 2012; Hoyer et al., 2008). For multi-structure data, it was initially treated as multiple single-structure problems (Tank et al., 2021; Peters et al., 2017). However, this leads to a situation where a new model needs to be refitted whenever a new causal structure appears.

Recently, to address the problem of multi-structure scenarios, existing methods typically employ amortized causal discovery approaches (Lorch et al., 2022b; Löwe et al., 2022; Huang

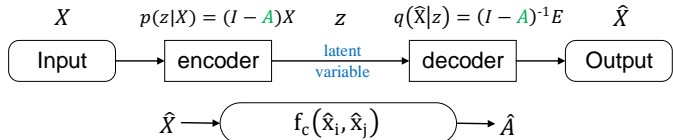

Figure 5: Generative framework to solve single-structure and complex-variable data.

et al., 2020a; Dhir and Lee, 2020; Huang et al., 2020b, 2019). Specifically, multiple causal structures are viewed as a distribution, with each structure being a sample within it. An estimated adjacency matrix of the structure, $\hat{A}$ ($\hat{A}_{i,j} = \hat{\mathcal{E}}_{i,j}$), can be obtained through the generative model shown in Figure 4, and then the causal representation $\hat{X}$ is derived via the SCM. As simple variables, $\hat{X}$ is equivalent to $X$, allowing the reconstruction loss to be formulated as distance measure between $\hat{X}$ and $X$, i.e., $E_{q(\hat{\mathcal{E}}_{i,j}|z)}[log p(z|(x_i, x_j))] = distance(\hat{X}, X)$.

### 4.1.2 CAUSAL REPRESENTATION LEARNING

In causal representation learning, we only consider those cases involving complex variables, i.e., situations where $\hat{X} \neq X$. In these cases, it is usually accompanied by a single, fixed, and known[6] causal structure (Fan et al., 2022; Wu et al., 2022; Lv et al., 2022; Jiang and Veitch, 2022), i.e., in input-output framework, causal structure does not need to be outputted. Under such an assumption, as shown in Figure 5, the noise term $E$ of the SCM can be treated as a latent variable, thus the encoder and decoder can be viewed as a back-and-forth mapping between $X$ ($\hat{X}$) and $E$ through known adjacency matrix $A$ ($\mathcal{G}$). Finally, the corresponding $\hat{A}$ is obtained from $\hat{X}$ through causal classifier $f_c$. Therefore, the reconstruction loss can be measured by the distance between $\hat{A}$ and the known structure $A$, that is, $E_{q(\hat{X}|z)}(log p(z|X)) = distance(\hat{A}, A)$.

### 4.1.3 MODEL GAP

In summary, when conducting structural learning on multi-structure data, simple variables need to provide $\hat{X} = X$ to constrain the optimization process. Similarly, when conducting representation learning on complex variables, a single fixed structure needs to provide a known $A$ to constrain the optimization process. Simply put, multi-structure data relies on the assumption of simple variables, while complex variables depend on the assumption of a single structure. The causal discovery of indefinite data needs to simultaneously handle multi-structure and complex variables, therefore, the model frameworks of causal structure learning and causal representation learning cannot be directly applied to the causal discovery task of indefinite data.

Existing work on indefinite data corroborates this point. For instance, Ke et al. (2021), despite the causal data being multi-structured and complex variables, it can only be applied to very simple toy models of causal mechanisms, such as physical force models. Similarly, Li et al. (2020), although it also considers the causal relationships of indefinite data, it

---

6. "Known" means that the causal structure $\mathcal{G}$ can be incorporated into the computation as a known quantity, both during the training and validation processes.

requires long-term observation data for each variable to determine the causal relationship. These methods avoid the model gap either by simplifying the causal pattern or significantly increasing the sample information.

## 4.2 Fundamental Framework

We propose a causal model suitable for indefinite data by combining the models of structural learning and representation learning, which can simultaneously discover the causal structure and causal representation of indefinite data. We first estimate the causal structure, referred to as $\hat{A}^s$, according to the framework of structural learning, and then generate representations according to the framework of representation learning, treating $\hat{A}^s$ as the known $A$.

Considering the latent confounders, the SCM is written as:

$$x_{m,j} = \sum_{x_{m,i} \in Pa(x_{m,j})} f_{m,ij} x_{m,i} + \sum_{l_{m,k} \in Ec(x_{m,j})} g_{m,kj} l_{m,k} + \epsilon_{x_{m,j}} \tag{2}$$

where $Pa(x_{m,j})$ represents the parent set of $x_{m,j}$, $Ec(x_{m,j})$ is the confounder set having effects on $x_{m,j}$, $f$ and $g$ denote the causal saliency [7] and confounding strengths, respectively, $\epsilon$ represents the exogenous i.i.d., noise term, and $K$ is the number of latent confounders. $x \in \mathbb{R}^{N \times D}, l \in \mathbb{R}^{K \times D}, \epsilon_{x_{s,j}} \in \mathbb{R}^{N \times D}, 0 \le i \ne j < N, 0 \le k < K$. The matrix form reads:

$$X = AX + BL + E \tag{3}$$

We design a couple of encoder and decoder to model the generating process of causal representation:

$$Encoder : W = f_1(X) \tag{4}$$

$$Decoder : \hat{X}^* = f_2(W(BL + E)) \tag{5}$$

where $f(\cdot)$ perform nonlinear transforms (neural network as GNN or MLP layers are popular choices) and $W$ represent $(I - A)^{-1}$. Please note, $f_1(X)$ is an abbreviation of $f_1(X(BL + E)^{-1})$ as $X$ consist of $BL + E$ w.r.t. $W$. Decoder can be written by a maximization of leg-evidence:

$$\frac{1}{M} \frac{1}{S} \sum_{m=1}^{M} \sum_{s=1}^{S} \log p(X_{s,m}) = \frac{1}{M} \frac{1}{S} \sum_{m=1}^{M} \sum_{s=1}^{S} \log \int p(X_{s,m}|W) p(W) dW \tag{6}$$

Continuing the theory of variational Bayes, we regard $W$ as the latent variable in variational autoencoder (VAE) (Kingma and Welling, 2022) and use variational posterior $q(W|X)$ to approximate the intractable posterior $p(W|X)$, thus the evidence lower bound (ELBO) reads:

---

7. causal saliency is defined as the probability of this edge being a causal relationship. In this paper, we set the causal saliency of each edge to be within [0, 1]. For example, a causal saliency of $f_{i,j} = 0.6$ indicates a 60% probability of a causal relationship existing from node $i$ to node $j$.

$$\mathcal{L}_{ELBO}^{s,m} = -KL(q(W|X_{s,m})||p(W)) + E_{q(W|X_{s,m})}[\log p(X_{s,m}|W)] \tag{7}$$

For simplicity, we model the prior as the standard normal $p(W) = \mathcal{MN}_{N \times N}(0, I, I)$, which indicates that each causal saliency $p(f_{ij}) = \mathcal{N}(0, 1)$. Note that even though the nodes are probably connected in a true graph, however, they are independent in prior.

In the causal view, our framework consists of two functions: a causal strength encoder: $\mathcal{X} \to \mathcal{G}$ and a causal representation decoder: $\mathcal{G} \to \widehat{\mathcal{X}}$.

## 4.3 Estimation of Confounding Effect

We use $c_{m,j} = \sum_{l_{m,k}} g_{m,kj} l_{m,k}$ to describe the confounding effect on $\widehat{x}_{m,j}$ and $C = BL$ to describe the corresponding matrix form. Inspired by (Agrawal et al., 2021), we proposed an estimation about $C$ (See more details from Appendix D):

$$c_{m,j} = \frac{p(x_{m,j})p(L|x_{m,j})}{\sum_i^N p(x_{m,i})p(L|x_{m,i})} x_{m,j} \tag{8}$$

Equation 8 only works when the expectation $E_{p(X)}(X|L)$ is much greater than the expectation $E_{p(X)}(X|\epsilon)$. It collaborates with the inductive bias that when confounding effects drastically exceed independent noise, $X$ is approximately contributed by $C$ rather than $E$. Therefore, the disentangled causal representation $\hat{X}$ can be written as:

$$\hat{X} \approx \begin{cases} \hat{X}^* - C, & (E_{p(X)}(X|L) \gg E_{p(X)}(X|\epsilon)) \\ \hat{X}^*, & else \end{cases} \tag{9}$$

## 4.4 Explanation

### 4.4.1 HOW TO EXTEND TO COMPLEX VARIABLES?

Given the existence of deconfoundment, we can, without loss of generality, write the SCM as: $\hat{x}_j = \sum_{\hat{x}_i \in Pa(x_j)} f_{ij} \hat{x}_i + \epsilon_{x_j}$, where the independence of $\epsilon$ ensures the Causation Condition. That is, we can directly recover the causal relationship from the causal representation $\hat{x}$. For instance, we can use causal classifier $f_c$ mentioned in Section 2.2. Additionally, if we linearly make causal representations $\hat{a}$ to fit $\hat{b}$ with a learnable parameter $k$ in a downstream task, and obtain the corresponding residuals: $\Sigma_{\hat{b}} = \hat{b} - k\hat{a}, \Sigma_{\hat{a}} = \hat{a} - \frac{1}{k}\hat{b}$. Then, different causal relations can be determined through the independence combination between residuals and representations:

- $\Sigma_{\hat{a}} \perp\!\!\!\perp \hat{b}, \Sigma_{\hat{b}} \not\!\perp\!\!\!\perp \hat{a} \Rightarrow \hat{b} \to \hat{a}$

- $\Sigma_{\hat{a}} \not\!\perp\!\!\!\perp \hat{b}, \Sigma_{\hat{b}} \perp\!\!\!\perp \hat{a} \Rightarrow \hat{a} \to \hat{b}$

- $\Sigma_{\hat{a}} \not\!\perp\!\!\!\perp \hat{b}, \Sigma_{\hat{b}} \not\!\perp\!\!\!\perp \hat{a} \Rightarrow l \to \hat{a}, l \to \hat{b}$

- $\Sigma_{\hat{a}} \perp\!\!\!\perp \hat{b}, \Sigma_{\hat{b}} \perp\!\!\!\perp \hat{a} \Rightarrow \hat{a} \to l, \hat{b} \to l$

### 4.4.2 How to Extend to Multi-structure Data?

In contrast with popular methods that intuitively treat the noise matrix as a latent variable (Yu et al., 2019; Chen et al., 2023c) ($\mathcal{X} \rightarrow \mathcal{E}$ and $\mathcal{E} \rightarrow \widehat{\mathcal{X}}$), we attempt to regard the causal saliency as a latent variable, thereby enabling one model to learn multiple structures. From the overall view, sampling from a set of DAGs $\mathcal{G}_m = \{\mathcal{E}_m, \mathcal{V}_m\}_{m=1}^M$ is equal to generate a set of causal saliencys which reads:

$$p(A) = \{p(A_m)\}_{m=1}^M \tag{10}$$

## 5 Baseline Metrics for Evaluation

In this section, we focus on evaluating the performance of our proposed probabilistic model in comparison with existing causal deep learning methods on indefinite data (i.e., *Causalogue* and *Causaction* datasets), guided by three key research questions:

- **Causal Structure Recovery (Q1):** How do existing causal discovery methods compare to ours in terms of recovering the underlying causal structure in the context of indefinite data?

- **Causal Representation Learning (Q2):** To what extent do the representations learned by existing methods and our approach encode causal relationships when trained on indefinite data?

- **Disentangling Confounding Effects (Q3):** How effectively do current methods for addressing confounding disentangle confounding effects in indefinite data compared to our approach?

### 5.1 Existing Approaches and Details of Implementation

To the best of our knowledge, no existing approach can be applicable in indefinite data. So we choose the SOTA work in Causal Discovery from multi-structure data and complex variables, respectively.

In multi-structure data, we evaluate our model with ACD (Löwe et al., 2022) and AVICI (Lorch et al., 2022a). In complex variables, we evaluate our model with CAE (Chen et al., 2023c), CVAE (Chen et al., 2023b), and DAG-GNN (Yu et al., 2019). Meanwhile, for the disentanglement, we evaluate our model with some SOTA work focusing on latent confounders: **pcss** (Agrawal et al., 2021), **LFCM** (Squires et al., 2022), and **GIN** (Xie et al., 2020). In the experiment, we made some necessary modifications to these methods to adapt them to indefinite data. For example, for ACD and AVICI, we increased the dimensions of the hidden layers to enlarge the representation space, while mapping the reconstruction loss into the correlation relationship space. For those methods focusing on complex variables, we replaced the latent variables with causal saliency. Additionally, we provide mainstream pre-trained models as baseline performances. For the Causalogue dataset, we set RoBERTa-base[8] as the baseline model, and for the Causaction dataset,

---

8. https://huggingface.co/docs/transformers/model_doc/roberta

Table 5: Performance of recovering causal graphs.AUROC and MSE are continuous metrics, computed by converting the causal graph into an adjacency matrix and comparing it against the ground truth graph. For all results obtained by our model, we perform a Student's t-test to ensure that the performance improvement over other methods falls within a 95% confidence interval.

| Method | Causalogue | | | Causaction | | |
|---|---|---|---|---|---|---|
| | AUROC | MSE | HD | AUROC | MSE | HD |
| RoBERTa | $0.31_{\pm0.048}$ | $0.57_{\pm0.009}$ | $2.55_{\pm0.022}$ | - | - | - |
| VideoMAE | - | - | - | $0.51_{\pm0.029}$ | $0.58_{\pm0.021}$ | $2.4_{\pm0.034}$ |
| ACD | $0.55_{\pm0.024}$ | $0.31_{\pm0.005}$ | $0.82_{\pm0.018}$ | $0.65_{\pm0.011}$ | $0.41_{\pm0.013}$ | $1.4_{\pm0.023}$ |
| AVICI | $0.57_{\pm0.019}$ | $0.37_{\pm0.003}$ | $0.86_{\pm0.024}$ | $0.69_{\pm0.009}$ | $0.44_{\pm0.011}$ | $1.2_{\pm0.016}$ |
| CAE | $0.54_{\pm0.021}$ | $0.41_{\pm0.005}$ | $0.79_{\pm0.021}$ | $0.61_{\pm0.012}$ | $0.48_{\pm0.012}$ | $1.3_{\pm0.019}$ |
| CVAE | $0.56_{\pm0.014}$ | $0.40_{\pm0.001}$ | $0.88_{\pm0.013}$ | $0.59_{\pm0.011}$ | $0.51_{\pm0.015}$ | $1.6_{\pm0.021}$ |
| DAG-GNN | $0.41_{\pm0.034}$ | $0.36_{\pm0.003}$ | $0.78_{\pm0.015}$ | $0.59_{\pm0.007}$ | $0.45_{\pm0.009}$ | $1.8_{\pm0.020}$ |
| Ours | $\mathbf{0.69}_{\pm0.019}$ | $\mathbf{0.26}_{\pm0.002}$ | $\mathbf{0.49}_{\pm0.019}$ | $0.78_{\pm0.008}$ | $\mathbf{0.30}_{\pm0.009}$ | $\mathbf{1.1}_{\pm0.023}$ |
| Ours$_{notears}$ | $0.67_{\pm0.011}$ | $0.29_{\pm0.001}$ | $0.54_{\pm0.013}$ | $\mathbf{0.81}_{\pm0.005}$ | $0.33_{\pm0.008}$ | $1.1_{\pm0.018}$ |

we set videoMAE[9] as the baseline model. For these two baseline models, we froze their pre-training parameters and only trained the classification layer.

In our Experiments, we utilized RoBERTa-base as our pre-trained model for generating word embeddings as input in the *Causalogue*. Throughout the training process, a learning rate of 1e-5 was set, with the batch size and epochs set to 16 and 50, respectively. The dimension of the hidden layers within the network was also set to 768. For the *Causaction*, we use less batch size with 4 to overcome the variable length and adopt more dimensions of the hidden layers (1024) to match the more complex information in video representation. The entire training procedure was conducted on an NVIDIA GEFORCE RTX 3090 graphics processing unit. In both datasets, 100 samples were randomly selected for a valid set and 200 samples were randomly selected for a test set. Each result is evaluated by 10-fold cross-validation.

## 5.2 Causal Structure (Q1)

To answer Q1, we evaluated the performance of recovering causal structures (causal graphs) on *Causalogue* and *Causaction*, using 3 different metrics: area under a receiver operating characteristic (AUROC), mean Squared error (MSE), and Hamming distance (HD).

Table 5 shows that our baseline model significantly outperforms existing methods with the applied necessary modifications. We believe this is due to the excessive specific assumptions made by existing methods for certain forms of data, which hinder their extension to a broader range of data forms. For instance, with DAG-GNN, even though we modified latent variables to adapt to indefinite data, with the acyclic constraint from NOTEARS (Zheng et al., 2018), a unique phenomenon emerges during the optimization process: the adjacency matrix $A$ tends to make $A_{ij}$ and $A_{ji}$ identical. This is advantageous for traditional causal

---

9. `https://huggingface.co/docs/transformers/model_doc/videomae`

Table 6: Performance of recovering causal graphs out of distribution

| Method | *Causalogue* | | | *Causaction* | | |
|---|---|---|---|---|---|---|
| | AUROC | MSE | HD | AUROC | MSE | HD |
| RoBERTa | $0.22_{\pm 0.057}$ | $0.63_{\pm 0.025}$ | $2.67_{\pm 0.073}$ | - | - | - |
| VideoMAE | - | - | - | $0.45_{\pm 0.024}$ | $0.77_{\pm 0.046}$ | $4.4_{\pm 0.153}$ |
| ACD | $0.51_{\pm 0.031}$ | $0.46_{\pm 0.025}$ | $1.63_{\pm 0.049}$ | $0.53_{\pm 0.034}$ | $0.58_{\pm 0.028}$ | $2.1_{\pm 0.046}$ |
| AVICI | $0.51_{\pm 0.045}$ | $0.46_{\pm 0.032}$ | $1.13_{\pm 0.051}$ | $0.60_{\pm 0.049}$ | $0.54_{\pm 0.037}$ | $2.8_{\pm 0.041}$ |
| CAE | $0.46_{\pm 0.033}$ | $0.46_{\pm 0.035}$ | $1.37_{\pm 0.054}$ | $0.61_{\pm 0.035}$ | $0.63_{\pm 0.039}$ | $2.5_{\pm 0.055}$ |
| CVAE | $0.49_{\pm 0.044}$ | $0.48_{\pm 0.028}$ | $1.37_{\pm 0.046}$ | $0.52_{\pm 0.039}$ | $0.49_{\pm 0.045}$ | $2.9_{\pm 0.048}$ |
| DAG-GNN | $0.33_{\pm 0.041}$ | $0.43_{\pm 0.029}$ | $1.45_{\pm 0.049}$ | $0.48_{\pm 0.044}$ | $0.53_{\pm 0.039}$ | $2.7_{\pm 0.043}$ |
| Ours | $\mathbf{0.61}_{\pm 0.024}$ | $\mathbf{0.35}_{\pm 0.008}$ | $\mathbf{0.94}_{\pm 0.027}$ | $\mathbf{0.66}_{\pm 0.016}$ | $\mathbf{0.42}_{\pm 0.014}$ | $\mathbf{1.8}_{\pm 0.031}$ |
| Ours$_{notears}$ | $0.55_{\pm 0.019}$ | $0.39_{\pm 0.005}$ | $0.99_{\pm 0.023}$ | $\mathbf{0.61}_{\pm 0.015}$ | $0.43_{\pm 0.018}$ | $2.1_{\pm 0.028}$ |

data with unknown causal order, but conflicts with the linear order in indefinite data. Moreover, we found that methods for multi-structured data (ACD and AVICI) perform only second-best to our method. This confirms that structure and representation are two individual aspects: multi-structure data have common laws, regardless of whether they are in simple or complex variables.

In addition, the ability to generalize out of distributions is essential for multi-structure data. To test whether these models can maintain robustness when encountering new causal structures, we designed a simple 10-fold experiment. In each fold, we randomly selected 2 structures (of *Causalogue*) or processes (of *Causaction*) for the test set, with all its samples prohibited from appearing in the train and valid sets.

Table 6 records the results of the cross-distribution test. Our method consistently outperforms existing methods, and the entire statistical result shows a situation similar to that of Table 5. Additionally, we noticed that the standard deviation of our method is much lower than other methods. We consider that for the reason there are similarities among some structures in the dataset. For instance, in the *Causalogue* dataset, Hybrid_II is very similar to Hybrid_I, but significantly different from the other 8 structures. In the *Causaction* dataset, many common causal relationships exist among "friedegg" and "pancake". When these structures are chosen for the test set in certain folds, the model can find "answers" from similar structures in the train set. However, when similar structures are all present in the test set fold (e.g., the test set includes Hybrid_I and Hybrid_II), it is difficult for the trained structures to manifest apparent invariance. However, the lowest standard deviation once again demonstrated the superiority of our baseline in releasing many assumptions about data forms. In other words, existing methods tend to rely on specific hypotheses to recover causal relationships, while our approach is more inclined to let the model itself learn the causal relationships.

## 5.3 Causal Representation (Q2)

Evaluating causal representation is another crucial aspect of indefinite data. Causal representation can be evaluated on both correlation and causation[10]. Specifically, we assume $x_i$

---

10. We believe that causal representations with causal relationships should satisfy correlation (although the converse is not true), therefore correlation can serve as an auxiliary evaluation metric.

Table 7: Performance of learning causal representations

| Method | Causalogue | | | | Causaction | | | |
|---|---|---|---|---|---|---|---|---|
| | Cas | | Cor | | Cas | | Cor | |
| | AUROC | MSE | AUROC | MSE | AUROC | MSE | AUROC | MSE |
| RoBERTa | $0.42_{\pm0.031}$ | $0.76_{\pm0.136}$ | $0.84_{\pm0.022}$ | $0.49_{\pm0.031}$ | - | - | - | - |
| VideoMAE | - | - | - | - | $0.48_{\pm0.012}$ | $0.43_{\pm0.020}$ | $0.81_{\pm0.011}$ | $0.34_{\pm0.009}$ |
| ACD | $0.52_{\pm0.026}$ | $0.64_{\pm0.074}$ | $0.91_{\pm0.013}$ | $0.43_{\pm0.022}$ | $0.59_{\pm0.006}$ | $0.39_{\pm0.009}$ | $0.88_{\pm0.001}$ | $0.28_{\pm0.005}$ |
| AVICI | $0.57_{\pm0.021}$ | $0.59_{\pm0.032}$ | $0.91_{\pm0.017}$ | $0.31_{\pm0.016}$ | $0.62_{\pm0.002}$ | $0.34_{\pm0.009}$ | $0.94_{\pm0.001}$ | $0.21_{\pm0.001}$ |
| CAE | $0.61_{\pm0.023}$ | $0.52_{\pm0.047}$ | $0.93_{\pm0.011}$ | $0.32_{\pm0.013}$ | $0.64_{\pm0.001}$ | $0.36_{\pm0.011}$ | $0.92_{\pm0.003}$ | $0.25_{\pm0.003}$ |
| CVAE | $0.62_{\pm0.021}$ | $0.55_{\pm0.066}$ | $0.91_{\pm0.006}$ | $0.29_{\pm0.024}$ | $0.61_{\pm0.005}$ | $0.31_{\pm0.005}$ | $0.92_{\pm0.001}$ | $0.23_{\pm0.003}$ |
| DAG-GNN | $0.59_{\pm0.019}$ | $0.55_{\pm0.059}$ | $0.90_{\pm0.017}$ | $0.39_{\pm0.009}$ | $0.63_{\pm0.003}$ | $0.33_{\pm0.013}$ | $0.91_{\pm0.002}$ | $0.26_{\pm0.002}$ |
| Ours | $\mathbf{0.68}_{\pm0.016}$ | $\mathbf{0.43}_{\pm0.058}$ | $\mathbf{0.95}_{\pm0.008}$ | $\mathbf{0.26}_{\pm0.011}$ | $\mathbf{0.79}_{\pm0.005}$ | $\mathbf{0.26}_{\pm0.004}$ | $0.96_{\pm0.002}$ | $\mathbf{0.15}_{\pm0.001}$ |
| Ours$_{notears}$ | $0.65_{\pm0.009}$ | $\mathbf{0.42}_{\pm0.025}$ | $0.94_{\pm0.013}$ | $\mathbf{0.21}_{\pm0.005}$ | $0.73_{\pm0.008}$ | $0.31_{\pm0.008}$ | $0.95_{\pm0.002}$ | $0.19_{\pm0.001}$ |

and $x_j$ to be any two causal representations that need to be tested. We propose a correlation matrix, $Cor$, to verify the performance in correlation, where $Cor_{ij} = cossin(x_i, x_j)$. Moreover, we train a $f_c$ (introduced in Section 2.2) to extract causal relationships, $Cas_{ij} = f_c(x_i || x_j)$. Both $Cor$ and $Cas$ are evaluated by AUROC and MSE, to demonstrate the performance of the causal representation in correlation and causation, respectively.

Table 7 presents the performance in correlation and causation. The results suggest that the representation more easily grasps the information of correlation, while causation, an asymmetric and underlying relation poses a more challenging topic in representation learning. Moreover, our method significantly outperforms others, even when we have modified them to adapt complex variables. This reason aligns with Section 5.2, for instance, the causality in ACD is based on the Granger causality hypothesis in time series, which stresses the faithfulness of single clues to causation. However, when it is expanded to other types of data (like the current indefinite data), it is tough to ascertain the correct set of parent nodes for causal representation. In addition, similar to the performance of causal structure, methods of complex variables (CAE, CVAE, DAG-GNN) also show superior performance in Table 7 over the multi-structure data methods. Thus, we can emphasize that representation and structure are two separate dimensions, and the concurrent existence of complex variables and multi-structure data lead to new challenges.

## 5.4 Disentanglement (Q3)

We created a set of synthetic datasets to evaluate the estimation of confounding effects. Specifically, We randomly draw Causal DAG from a random graph model with an expected neighborhood size of 5 and consider graphs with the number of observed nodes $N \in \{20, 50, 100\}$. For probing how our approach is affected by the pervasiveness of confounding, we assume that each confounder $l_k$ is a direct cause of node $x_i$ with a chance $P \in \{0.1, 0.4, 0.7\}$. Given the graph, we stochastically set a trend type for each causal saliency weight with equal probability. Meanwhile, we add $N(0, \sigma_{noise}^2)$ noise to each node. Finally, we consider the number of confounders $K \in \{1, 5, 10\}$ and the number of samples of each skeleton $n \in \{5, 10, 50\}$, respectively.

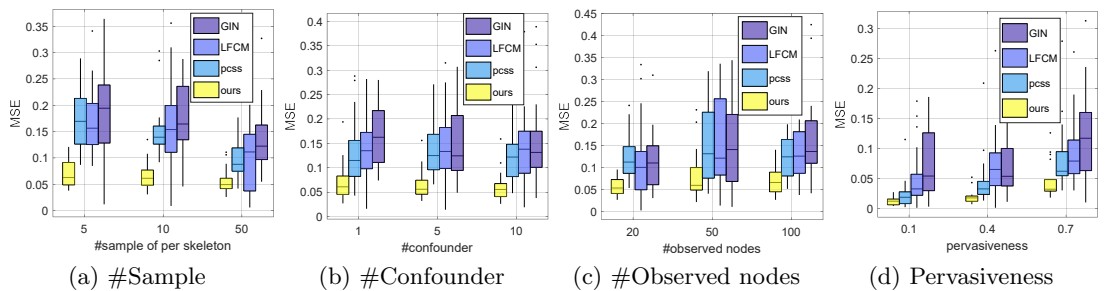

| (a) #Sample | (b) #Confounder | (c) #Observed nodes | (d) Pervasiveness |

Figure 6: MSE error across all ingredients setting for estimating $C$ via GIN, LFCM, pcss, and ours.

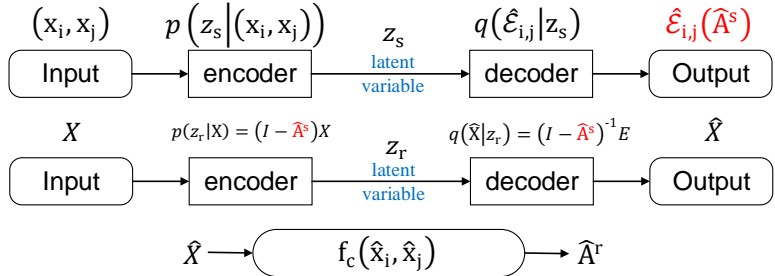

Figure 7: Generative framework for indefinite data.

In Figure 6, we quantify the mean-squared estimation (MSE) error of $C$. Our method likewise performs best in all ingredient settings, demonstrating that our confounding disentanglement pools the statistical strength better than other estimation algorithms in multi-structure data. Besides, combined with the conclusion in (Agrawal et al., 2021), this error should decrease as the number of samples $n$ increases. Figure 6 (a) is exactly indicative of this conclusion.

## 6 Discussion

### 6.1 Summary

This paper focuses on causal inference for a novel paradigm of data - indefinite data, characterized by multi-structure data and complex variables. These two features differ greatly from traditional experimental data, thereby causing existing methods to be non-adaptive to indefinite data. To provide a good starting point for causal research on indefinite data, we introduce two brand new datasets- *Causalogue* and *Causaction*, analyze the challenges brought about by the coexistence of multi-structure data and complex variables, and propose a corresponding probabilistic framework. In the experiments, we exhibit benchmark results for both structure and representation and share intrinsic insights in the extension of disentanglement.

Table 8: Performance among different pre-trained models

| model | Causalogue | | | | model | Causaction | | | |
|---|---|---|---|---|---|---|---|---|---|
| | Structure | | Representation | | | Structure | | Representation | |
| | AUROC | MSE | AUROC | MSE | | AUROC | MSE | AUROC | MSE |
| RoBERTa | 0.69 | 0.26 | 0.68 | 0.43 | I3D | 0.78 | 0.30 | 0.79 | 0.26 |
| BERT | 0.67 | 0.36 | 0.66 | 0.41 | Timesformer | 0.81 | 0.33 | 0.81 | 0.28 |
| XLM | 0.68 | 0.28 | 0.69 | 0.44 | videoMAE | 0.85 | 0.38 | 0.84 | 0.29 |
| XLNet | 0.69 | 0.27 | 0.67 | 0.45 | - | - | - | - | - |

## 6.2 Inconsistency

However, such a probabilistic framework is inconsistent with the learning of causal structure and causal representation. As shown in Figure 7, the estimated structure $\hat{A}^s$ is first obtained through the multi-structure generative model, and then input into the representation generative model to obtain the causal representation. Another estimated structure $\hat{A}^r$ is obtained from $\hat{X}$ through a causal classifier $f$. There are two reconstruction losses, calculated by measuring the distance between the two estimated structures and the ground truth $A$, i.e., $E_{q(\hat{\mathcal{E}}_{i,j}|z_s)}[logp(z_s|(x_i,x_j))] = distance(\hat{A}^s, A)$, $E_{q(\hat{X}|z_r)}(logp(z_r|X)) = distance(\hat{A}^s, A)$. Therefore, significant inconsistency between $\hat{A}^r$ and $\hat{A}^s$ arises due to the independent backpropagation from $\hat{A}^s$ and $\hat{A}^r$ initiated by the two reconstruction losses.

We have elaborated on this problem and proposed an intervention-based improvement in our latest work (Chen et al., 2024). Broadly speaking, we attempt to apply various interventions $do(x_1)$, $do(x_2)$, etc. to $\hat{X}$, resulting in the intervened adjacency matrices $\hat{A}^r_{do(x_1)}$, $\hat{A}^r_{do(x_2)}$, etc. Concurrently, corresponding interventions are performed on $\hat{A}^S$ to obtain $\hat{A}^s_{do(x_1)}$, $\hat{A}^s_{do(x_2)}$, etc. According to the principle of causal abstraction (Beckers and Halpern, 2019), if $\hat{A}^r_{do(x_1)}$, $\hat{A}^r_{do(x_2)}$, ... correspondingly equal $\hat{A}^s_{do(x_1)}$, $\hat{A}^s_{do(x_2)}$, ..., the causal model of $\hat{X}$ and $\hat{A}^s$ is also equivalent.

## 6.3 Pre-trained Model

Since complex variables do not exist in numerical form, we use the representations obtained from pre-trained models as input. For instance, in the input-output framework of Causalogue, we default to using the pre-trained representations from RoBERTa-base as $X$, and in Causaction, we use the pre-trained representations from I3D. However, whether the initial representations provided by different pre-trained models will have a significant impact on representation learning and structural learning is an unconfirmed issue. Therefore, we have selected some competitive pre-trained models: Bert (Devlin et al., 2019), Roberta (Liu et al., 2019), XLM (Lample and Conneau, 2019), XLNet (Yang et al., 2020) for the Causalogue dataset, and I3D (Carreira and Zisserman, 2017), Timesformer (Bertasius et al., 2021), videoMAE (Tong et al., 2022) for the Causaction dataset.

Table 8 shows the differences between different pre-trained models, where the text-based pre-trained models (see the Causalogue column) do not cause significant changes, while the video-based pre-trained models (see the Causaction column) do. We speculate that this is because the dataset sizes involved in these video pre-training models have significantly

Table 9: Performance of generalization of CausalDialogue dataset.

| Method | Trained with Causalogue | | | Trained with CausalDialogue | | |
|---|---|---|---|---|---|---|
| | AUROC | MSE | HD | AUROC | MSE | HD |
| RoBERTa | $0.19_{\pm 0.056}$ | $0.72_{\pm 0.033}$ | $3.11_{\pm 0.073}$ | $0.28_{\pm 0.043}$ | $0.53_{\pm 0.024}$ | $2.15_{\pm 0.055}$ |
| ACD | $0.45_{\pm 0.022}$ | $0.51_{\pm 0.043}$ | $1.73_{\pm 0.055}$ | $0.58_{\pm 0.025}$ | $0.37_{\pm 0.022}$ | $1.4_{\pm 0.047}$ |
| AVICI | $0.48_{\pm 0.031}$ | $0.43_{\pm 0.038}$ | $1.25_{\pm 0.042}$ | $0.62_{\pm 0.044}$ | $0.29_{\pm 0.027}$ | $0.9_{\pm 0.033}$ |
| CAE | $0.52_{\pm 0.042}$ | $0.50_{\pm 0.029}$ | $1.67_{\pm 0.042}$ | $0.58_{\pm 0.035}$ | $0.31_{\pm 0.025}$ | $1.1_{\pm 0.042}$ |
| CVAE | $0.46_{\pm 0.038}$ | $0.44_{\pm 0.042}$ | $1.42_{\pm 0.031}$ | $0.59_{\pm 0.032}$ | $0.29_{\pm 0.036}$ | $1.0_{\pm 0.031}$ |
| DAG-GNN | $0.39_{\pm 0.034}$ | $0.49_{\pm 0.055}$ | $1.57_{\pm 0.034}$ | $0.63_{\pm 0.062}$ | $0.21_{\pm 0.025}$ | $1.1_{\pm 0.029}$ |
| Ours | $\mathbf{0.56}_{\pm 0.031}$ | $\mathbf{0.41}_{\pm 0.006}$ | $\mathbf{1.15}_{\pm 0.025}$ | $\mathbf{0.71}_{\pm 0.019}$ | $\mathbf{0.19}_{\pm 0.011}$ | $\mathbf{0.9}_{\pm 0.011}$ |

increased, while in the text-based pre-training models, the dataset sizes involved do not show significant differences. This may also suggest that a large pre-training dataset can provide richer context or domain information to the initial representation, which is beneficial for causal structure learning and representation learning.

## 6.4 Generalization

### 6.4.1 Generalization in Other Datasets

We will use CausalDialogue dataset as a test set for generalization to evaluate the performance of our proposed baseline, which is trained on the Causalogue dataset. This experiment has two main objectives. First, by discovering causal relationships in CausalDialogue, we aim to demonstrate that the texts generated by Causalogue based on a pre-defined template do not suffer from the "teaching to the test" risk. Instead, they contain genuine causal relationships that can generalize to other causal dialogue datasets, similar to the function of a style transfer experiment. Second, we seek to prove the generalization capability of indefinite data further.

Table 9 presents the results of testing on the CausalDialogue test set when the training sets are Causalogue and CausalDialogue, respectively. It is evident that the model trained on Causalogue is able to identify causal relationships in CausalDialogue. This indicates that the causal dialogues generated by Causalogue using pre-defined templates do not contain shortcuts or biases in their causal properties. Furthermore, the fact that the model trained on CausalDialogue performs better on the CausalDialogue test set demonstrates the promising generalization capability of our baseline on other indefinite datasets.

### 6.4.2 Generalization in Wild

Although this paper proposes benchmark datasets and baseline models for causal discovery in indefinite data, there are significant limitations in exploring indefinite data in wild environments. For example, the causal structures in the Causalogue and Causaction datasets only have some fixed types, while in the real world, there are far more types of causal structures. Additionally, in real-world data, the sparsity of causal relationships and the amount of causal variables can also vary widely. These hinder our ability to "discover anything", and suggest that the successful self-supervision and Next token prediction mechanisms in large language models may be key to applicability in wild environments.

Therefore, causal research on indefinite data will be a long-term topic. The ultimate goal of multi-structure data is to exhibit cross-distribution learning capability, while complex variables will eventually rely on a model-driven learning process. In previous work, we posited such a challenge, and this paper makes it feasible from a set of research basis. We hope this will attract an increasing number of researchers to contribute to the growing body of work applying causal inference to the real world.

### 6.5 Counterfactual Verification

In Section 3.3.2, we defined the causal annotation criteria for the *Causaction* dataset. Briefly, a causal relation $A \rightarrow B$ is considered to exist if the following two conditions are met: (1) "if A happens, then B happens" and (2) "if B happened, would A not have happened?" For convenience, we refer to this formulation as **counterarrow**. However, as pointed out by reviewer W8C2, an alternative criterion is the **counterfactual** formulation, which identifies causality based on the conditions: "if A happens, then B happens" and "if A did not happen, would B have happened?"

Both methods aim to infer causality from observed correlations. Specifically, the common component — "if A happens, then B happens" — merely indicates a correlation between A and B. Counterarrow seeks to identify whether the correlation is directional by testing whether "if B happened, would A not have happened?" since correlation is directionless whereas causality is inherently unidirectional. In contrast, counterfactuals introduces an intervention on A through "if A did not happen, would B have happened?" probing whether changes in A induce changes in B — a hallmark of causal influence that does not hold under mere correlation.

Each method, however, has limitations in specific contexts. For example, in the case of long causal chains (there are many intermediators in path A to B), counterarrow may fail, as it becomes difficult to ensure that B occurs without any subsequent occurrence of A. On the other hand, counterfactuals struggle in fork structures, particularly those involving multiple causes: unless a precise quantitative analysis of probability of B is conducted, it is often hard to detect a significant influence of A on B.

The Table 10 presents several example relations, along with the proportion of counterarrow and counterfactual judgments observed in the raw resource:

The Table 10 clearly illustrates that, for these examples, counterarrow tends to exhibit more pronounced distinctions in frequency relative to the raw data. This can be attributed to the fact that long-chain structures are rare in *Causaction*, whereas fork patterns involving multiple causes are common. Consequently, while interventions on A may lead to a reduction in the probability of B, the effect may not be sufficiently pronounced for annotators to consistently detect. Given this, we ultimately recommend the counterarrow method for annotation purposes.

## 7 Broader Impact Statement

The intended uses of our proposed datasets (*Causalogue* and *Causaction*) and baseline models are for scientific causal knowledge discoveries such as reasoning tasks in conversation, complex causes extraction in graphs and causal representation learning without limited assumptions. The datasets consist of generated dialogs, and videos and do not include any

Table 10: Counterarrow vs. counterfactual in *Causaction*. We show the frequences of each type of relations in the raw resource. Concretely, we identified two actions to serve as variables A and B. We define original as the proportion of samples in which "A occurs before B"; counterarrow as the proportion in which "B occurs before A"; and counterfactual as the proportion in which "B occurs without A occurring." The selected action pairs are those for which a causal relationship was annotated with over 95% agreement among annotators.

| A | B | Original | Counterarrow | Counterfactual |
|---|---|---|---|---|
| pour milk | stir | 42.3% | 0 % | 15.4% |
| crack eggs | fry | 31.5 % | 1.3 % | 12.3% |
| cut fruit | put in bowl | 22.4% | 4.1 % | 9.7 % |
| butter | cover toast | 19.6 % | 0 % | 1.4 % |
| turn on gas | fry | 52.4% | 0 % | 0 % |

social/personal information. We believe that a potential positive societal consequence of this work is that our two indefinite datasets and baseline model will help non-causality experts choose which causal data paradigms and frameworks they want to apply to their problem for causal discovery. The proposed datasets are more realistic for discussing causal relationships than existing causal datasets as 1) our proposed datasets satisfy both complex variables and multi-structure data, to our best knowledge, they are the first two datasets without structure and representation limitations 2) we introduced a new baseline model to the proposed datasets to discuss the causal relationships (causal structures and causal representations) and confounders of indefinite data and 3) our benchmark experiments show that the proposed baseline model is more aligned with indefinite causal relationships than existing algorithms for causal discovery.

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

## Appendix A. Effectiveness of $f_c$

To evaluate whether $f_c(\cdot)$ can learn the ability to discern causal relationships, we selected two datasets with simple variables for verification (complex variable causal representations do not have a fixed ground truth): Arrhythmia (Guvenir et al., 1997) and Netsim (Smith et al., 2011). During testing, we (1) randomly selected 100 pairs of variables with causal relationships and 100 pairs without causal relationships ("normal" in Table), (2)randomly selected 100 pairs of variables with causal relationships and reverse order of them as other 100 pairs ("reverse" in Table), (3)randomly selected 100 pairs of variables with causal relationships and 100 pairs randomly selected ("random" in Table). In the training setup, we designed $f_c(\cdot)$ as a two-layer MLP with a hidden state of 256, set the batch size to 32, the learning rate to 0.0006, and the number of epochs to 20. The F1 score of $f_c(\cdot)$ is shown in the Table 11.

<table>
<tr><td colspan="4">Table 11: F1 of $f_c(\cdot)$ in simple variables</td></tr>
<tr><th>Datasets</th><th>normal</th><th>reverse</th><th>random</th></tr>
<tr><td>Arrhythmia</td><td>0.97</td><td>0.98</td><td>0.99</td></tr>
<tr><td>Net-fmri</td><td>0.91</td><td>0.88</td><td>0.95</td></tr>
</table>

<table>
<tr><td colspan="4">Table 12: F1 of $f_c(\cdot)$ without context.</td></tr>
<tr><th>Datasets</th><th>normal</th><th>reverse</th><th>random</th></tr>
<tr><td>Causalogue</td><td>↓ 0.07</td><td>↓ 0.06</td><td>↓ 0.03</td></tr>
<tr><td>Causaction</td><td>↓ 0.04</td><td>↓ 0.05</td><td>↓ 0.04</td></tr>
</table>

Clearly, $f_c(\cdot)$ can distinctly reflect the causal relationships between data samples. In other words, as long as the correct causal relationship (rather than correlation) exists in the representation, $f_c(\cdot)$ has the ability to identify the specific direction and existence of the causal relationship.

To assess the performance of $f_c$ with complex variables, we conducted a similar ablation study. Since there is no direct ground truth for complex variables, we evaluated whether $f_c$ could maintain a certain level of recognition when irrelevant causal representations—which we refer to as "context" (e.g., the representations of unrelated utterances in the Causalogue dataset)—were removed. Through this experiment on invariance to context distribution shifts, we aimed to demonstrate that $f_c$ does not classify causal relationships by recognizing causally irrelevant context. Table 12 shows the difference in F1 scores for $f_c$ between scenarios with and without context. The results clearly indicate that the performance of $f_c$ did not significantly degrade after the removal of causally irrelevant context, which also eliminates the risk of circularity.

## Appendix B. Acyclicity Constraint

In this work, we eliminate the possibility of cyclic causal graphs by Assumption 2. However, to assess broader applicability, we seek to examine the performance of baseline models on datasets that do not adhere to this assumption. For such cases, inspired by Zheng et al. (2018), we propose an alternative constraint that is more practical for real-world implementation.

Let $A \in \mathcal{R}^{m \times m}$ be the (possibly negatively) weighted adjacency matrix of a directed graph. For any $\alpha > 0$, the graph is acyclic if and only if

$$tr[(I + \alpha A \circ A)^m] - m = 0 \tag{11}$$

Table 13: Comparisons between causal datasets

| Source | 4 variables | 5 variables | 6 variables |
|--------|-------------|-------------|-------------|
| LLM | 0.92 | 0.61 | 0.44 |
| Wild | 0.57 | 0.31 | 0.16 |

In practice, $\alpha$ could be set as a hyperparameter and its value depends on an estimation of the largest eigenvalue of $A \circ A$ in magnitude. This value is the spectral radius of $A \circ A$, and because of nonnegativity, it is bounded by the maximum row sum according to the Perron–Frobenius theorem. Hence, we use Equation 11 as the equality constraint when maximizing the ELBO. The learning problem is

$$\min_{A,\theta} f(A,\theta) \equiv -\mathcal{L}_{ELBO}$$
$$s.t.\ h(A) \equiv tr[(I + \alpha A \circ A)^m] - m = 0 \tag{12}$$

In the table **??**, we compare model performance under two different constraints: adherence to Assumption 2 and the Acyclicity Constraint. The results clearly demonstrate that our proposed probabilistic model retains effective causal identifiability even when Assumption 2 is not enforced.

## Appendix C. Aggrements between Different Dataset Scaling

In this section, we present a preliminary evaluation of the data scale for *Causalogue*. The evaluation focuses on two key factors: the number of causal variables and the source of the text (i.e., LLM-generated versus in-the-wild dialogues). The specific combinations of these factors are as follows:

- **4 variables + wild:** 200 samples, each consisting of 4 dialogue turns extracted from the RECCON dataset, with each sample containing at least one causal span.

- **5 variables + wild:** 200 samples, each consisting of 5 dialogue turns from RECCON, each containing at least one causal span.

- **6 variables + wild:** 200 samples, each consisting of 6 dialogue turns from RECCON, each containing at least one causal span.

- **4 variables + LLM:** 200 causally annotated samples generated by GPT-4 following the procedure described in Section 3.2.2, each containing 4 variables.

- **5 variables + LLM:** 200 GPT-4-generated causal samples with 5 variables.

- **6 variables + LLM:** 200 GPT-4-generated causal samples with 6 variables.

We employed a team of expert annotators to manually label the causal relationships in each sample. The agreement between the labels was then assessed using Cohen's kappa with respect to the ground truth labels. The specific agreement values are shown in the table 13.

It is evident that wild data fails to meet the consistency standards typically expected of benchmark datasets. We attribute this to the inherently subtle and ambiguous nature

of causal relationships in real-world dialogues. In contrast, LLM-generated samples exhibit higher annotation agreement due to the presence of clearly defined, human-specified causal structures. Additionally, as the number of variables increases, the complexity of the underlying causal graphs also increases, which in turn leads to decreased annotation consistency.

Given that four variables are sufficient to capture the fundamental building blocks of causal structure—namely, chains, forks, and colliders—and considering that research on indefinite data remains in its early stages, our primary goal is to provide a clean, mathematically tractable dataset. Work on scaling to more complex, real-world data distributions remains an important direction for future research, which will require more advanced data collection and annotation strategies.

## Appendix D. Confounding Disentanglement

Equations 2 and 3 indicate that latent confounders are a critical problem in our research: when they exist, non-autoregression SEM invalidates the VAE. Hence, to eliminate the effect of confounders and reconstruct the true causal relations, we consider the following disentanglement model in this paper:

$$\mathcal{H} = \mathcal{O} \cup \mathcal{C} \tag{13}$$

where $\mathcal{H} = \{(I-A)^{-1}(BL+E), \mathcal{E}_{\mathcal{H}}\}$, $\mathcal{O} = \{(I-A)^{-1}E, \mathcal{E}_{\mathcal{O}}\}$, $\mathcal{C} = \{(I-A)^{-1}BL, \mathcal{E}_{\mathcal{C}}\}$. From a causal graph view, graph $\mathcal{O} = \{X, \mathcal{E}_{\mathcal{O}}\}$, $\mathcal{E}_{\mathcal{O}}$ represents the edge '$x_i \rightarrow x_j$', and graph $\mathcal{C} = \{X \cup L, \mathcal{E}_{\mathcal{C}}\}$. $\mathcal{E}_{\mathcal{C}}$ represents the edge '$l_k \rightarrow x_j$'. Graph $\mathcal{H}$ is the full causal graph with all observed and latent relations. Note that $\mathcal{H}$ is only in theory because we can not obtain confounders. $\mathcal{H} = \mathcal{G}$ if we omit the latent confounders, which embrace the traditional causal discovery aspect (Equation 4 in Appendix B). $\mathcal{H} = \mathcal{O}$ if there are no confounders in this causal skeleton. See also Figure 8 for an illustration.

In general, the ultimate goal of causal discovery under confounding is to correct $\mathcal{G}$ and recover $\mathcal{H}$ from observed variables set $X$. However, indefinite data, such complicated data paradigm, makes it stubborn to achieve $\mathcal{H}$ because some prevalent assumptions about solving confounding can not hold here (e.g., there is no relation between any two observed variables Squires et al. (2022), or all observed variables are affected by one confounder Agrawal et al. (2021)), which results in the problems that we can not know and assume the locations, numbers, and effects of confounders.

To this end, we design this causal disentanglement, which makes relations of observed variables amenable without assuming confounders. For $\mathcal{O}$, we follow the variational interference in Figure 5, which can reconstruct the deconfounding variables $\widehat{X}$. For $\mathcal{C}$, we would like to compute the confounding effects $C_i$ on each observed variables $X_i$ instead of the value of confounders. The confounding effects $C \in \mathbb{R}^{N \times D}$ structurally resemble reconstruction variables $\widehat{X} \in \mathbb{R}^{N \times D}$. From the CAM view, it reads:

$$c_{m,j} = \sum_{l_{m,k} \in Ec(x_{m,j})} g_{m,kj} l_{m,k} (c \in \mathbb{R}^{N \times D}, l \in \mathbb{R}^{K \times D}, 0 \le i \ne j, k < N) \tag{14}$$

Due to the assumption of causal saliency is irrelevant to 'confounding strength' $g$, equation 14 is intractable without approximate statistics as we have shown in Equation 8, with

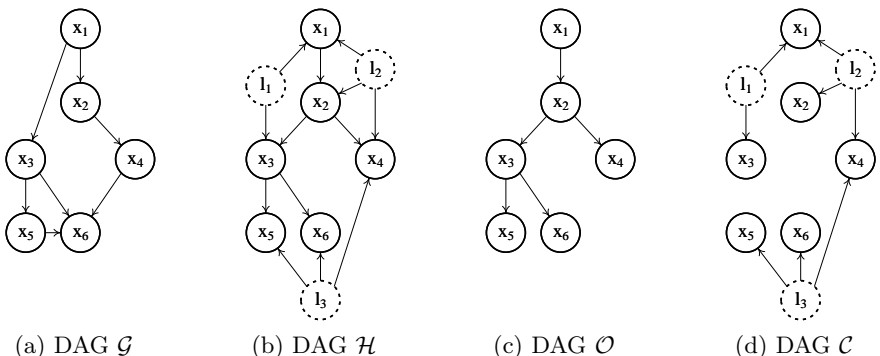

(a) DAG $\mathcal{G}$      (b) DAG $\mathcal{H}$      (c) DAG $\mathcal{O}$      (d) DAG $\mathcal{C}$

Figure 8: Four causal DAGs discussed in this paper. $\mathcal{G}$ represents the wrong causal structure lacking the consideration of latent confounders, $\mathcal{H}$ represents the true causal the structure given the whole observed variables and latent confounders, $\mathcal{O}$ and $\mathcal{C}$ are two subgraphs disentangled from $\mathcal{H}$. $\mathcal{O}$ has and only has relations between observed variables (i.e., $x_1 \rightarrow x_2$), and $\mathcal{C}$ has and only has relations from latent confounders to observed variables (i.e., $l_1 \rightarrow x_1$).

the following proofs: By equation 7 in the main text,

$$(x_j - C_j) = \sum_{x_i \in Pa(x_j)} A_{i,j}(x_i - C_i) + \epsilon_j \tag{15}$$

Hence,

$$C_j = \sum_{x_i \in Pa(x_j)} A_{i,j} C_i + B_j L \tag{16}$$

$$= \sum_{x_i \in Pa(x_j)} A_{i,j}[(I - A)^{-1}BL]_i + B_j L \tag{17}$$

Intuitively, the term $(I - A)^{-1}BL$ reflects a particular graph $\mathcal{Q}$ consisting of $p(X_s|L)$. Hence, we would like to transform Equation 17 into a statistic about '$p(X|L)$'. Fortunately, a sufficient statistic $C_j = E(X_j|L)$ is supported by Agrawal et al. (2021). It makes $\mathcal{Q}$ identifiable under the Gaussian $\epsilon$ and $E(\epsilon) = 0$. Along with their inference, we find the branch point under our data condition:

$$C_j = \sum_{x_i \in Pa(x_j)} A_{i,j}[(I - A)^{-1}BL]_i + B_j L$$

$$= E[\sum_{x_i \in Pa(x_j)} f_{ij}[(I - A)^{-1}E + (I - A)^{-1}BL]_i \tag{18}$$

$$+ B_j L | L]$$

If the $E(\epsilon)$ is negligible, we can obtain the same statistic due to $C_j = E[\sum_{x_i \in Pa(x_j)} f_{ij}[(I - A)^{-1}E + (I - A)^{-1}BL]_i + B_j L + \epsilon_j | L]$. In other words, when $E(L)$ apparently exceeds $E(\epsilon)$,

latent confounders essentially contribute the $C$, and naturally, $C$ is meaningless when it is mainly affected by $\epsilon$.

Finally, we could approximately estimate the discrete probability of $C$ under a strong confounding assumption:

$$C_j = E(x_j|L) \tag{19}$$

$$= \frac{\mathbb{P}(x_j)\mathbb{P}(L|x_j)}{\sum_i^N \mathbb{P}(x_i)\mathbb{P}(L|x_i)}x_j \tag{20}$$

Additionally, $C$ is not easy to calculate under weak confounding, so the dynamic reconstruction loss function does not consider C when confounding influence is insufficient (See Subsection "Dynamic Reconstruction Error" for details).

Equation 8 describes an expectation statistic irrelevant to $B$ while involved with $E$, which collaborates the inductive bias that when confounding effects drastically exceed independent noise, $X$ is approximately contributed by $L$ rather than $E$. We thus design a dynamic reconstruction loss $l_r$: when $l \gg \epsilon_x$ (i.e., the confounding effects are significant), $l_r$ measures the distance between $X$ and $\widehat{X} + C$; on the contrary, in the case that confounding effects are negligible, $l_r$ measures the distance between $X$ and $\widehat{X}$ as well as $C$ is hard to estimate.

## Appendix E. Sensitive Experiment of each baselines

In Section 5, we made some necessary modifications to the baseline methods so they could be applied to indefinite data. For instance, with ACD and AVICI, we increased the dimensions of the hidden layers to enlarge the representation space while mapping the reconstruction loss into the correlation relationship space. For methods that focus on complex variables, we replaced the latent variables with causal saliency. Therefore, we conducted sensitivity experiments to demonstrate that these modifications do not compromise the model's ability to learn causal representations or causal structures.

Figure 9 displays the differences between the modified methods (denoted by $*$, like ACD*) and their original counterparts on their respective evaluation experiments. Since the original evaluation datasets and metrics varied, we report the distance using the relative error:

$$\frac{\text{performance of modified model} - \text{performance of original model}}{\text{performance of original model}}$$

The results clearly indicate that although we made necessary changes to these methods, these modifications did not severely compromise their ability to learn causal representations and causal structures.

## Appendix F. Implementation Example

We formalized the dynamic variational inference model as follows: a causal saliency encoder $f_\varphi : \mathcal{X} \to \mathcal{G}$, an causal representation decoder $f_\theta : \mathcal{G} \to \widehat{\mathcal{X}}$, and an estimation function $f_\delta : \mathcal{X} \to \mathcal{C}$.

We resort to VAE to design the functions $f_\varphi$ and $f_\theta$ as shown in Figure 10. Specifically,

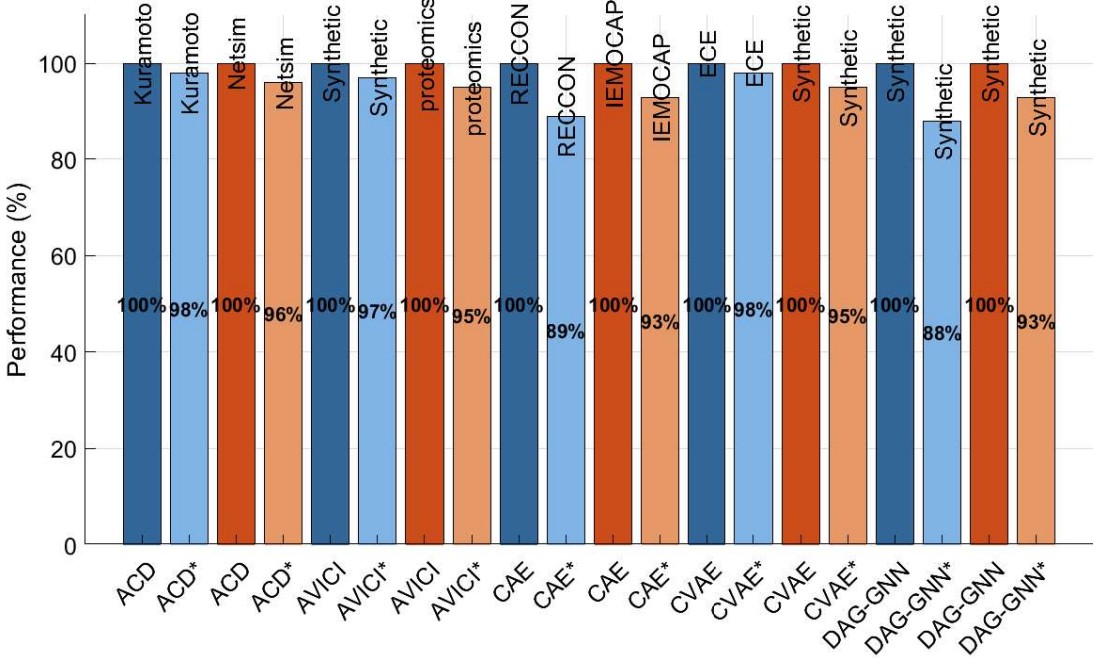

Figure 9: The performance of the modified model and the original model in the original evaluation datasets. We set the performance of the original model as 100%.

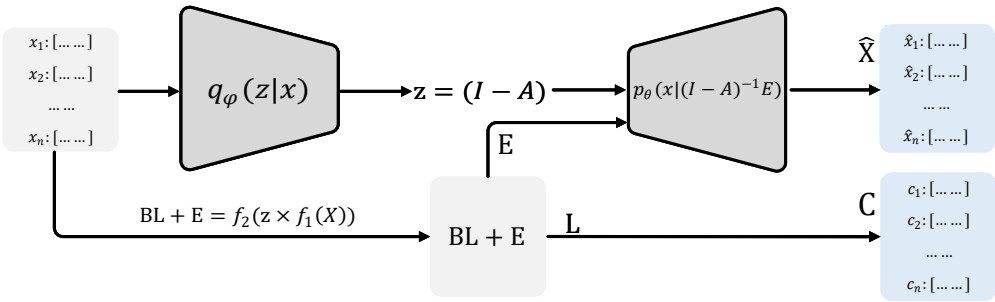

Figure 10: An implementation example of our framework. $q_{\varphi}(z|\mathcal{X})$ predicts the causal saliency from the input $X$. The predicted latent variable $z = (I - A)$, and then a causal representation decoder $p_{\theta}((x|(I - A)^{-1}E))$ learns to predict $\widehat{X}$ given the disentangled $E$ and inverse of predicted $z$.

### F.1 Encoder

The encoder $q_\varphi(z|\mathcal{X})$ applies a graph attention module $f_{att,\varphi}$ (Veličković et al., 2017) to the input. It produces an adjacent matrix across a lower triangular mask under Hypothesis 2.

$$q_\varphi(z|\mathcal{X}) = softmax(f_{att,\varphi}(X)) \tag{21}$$

The output $z$ implies the possible distribution of causal saliency over $\mathcal{X}$. Specifically, $z_{i,j} = 1$ indicates a high probability relation $x_j \to x_i$.

### F.2 Decoder

we extract $E$ by utilizing a multi-layer perceptron (MLP):

$$BL + E = GNN_{enc}(f_{att,\varphi}(X), X) \tag{22}$$
$$E = MLP_E(BL + E) \tag{23}$$

where $GNN_{enc}$ is instantiated by graph neural network: $GNN(\mathsf{A}, \mathsf{X}) = eLU(\mathsf{A} \times (\mathsf{X} \times \mathsf{W}))$, which yields a nonlinear multiple of adjacent matrix $\mathsf{A} \in \mathbb{R}^{N \times N}$, feature matrix $\mathsf{X} \in \mathbb{R}^{N \times D}$ and weight matrix $\mathsf{W} \in \mathbb{R}^{D \times H}$, where $H$ represents the dimensions of hidden layers. Then, the decoder accumulated the incoming messages to each node via causal saliency $z$ and employed a new graph neural network $GNN_{dec}$:

$$p_\theta((\hat{x}_c|z^{-1}E)) = GNN_{dec}(z^{-1}, E) \tag{24}$$

The output of the decoder $\hat{x}_c \in \mathbb{R}^{N \times D}$ equals the dimension of $\mathcal{X}$ and it is the pure causal representation of $\hat{x}$ without confounding.

### F.3 Confounding Estimation

We used the same MLP module to extract $L$ and two sigmoid functions: $\sigma_{p(x_j)}(\cdot)$ and $\sigma_{p(L|x_j)}(\cdot)$, to project $p(x_j)$ and $p(L|x_j)$ into the range of $(0,1)$, which expresses the probability estimating $c_j$.

$$L = MLP_L(BL + E) \tag{25}$$
$$c_j = \frac{\sigma_{p(x_j)}(x_j)\sigma_{p(L|x_j)}(L|x_j)}{\sum_i^N \sigma_{p(x_i)}(x_i)\sigma_{p(L|x_i)}(L|x_i)} x_j \tag{26}$$

The output $C \in \mathbb{R}^{N \times D}$ of Estimation module equals the individual-specific effects of confounding on each $x_j$ if there exactly exists strong confounding.

### F.4 Reconstruction Error

Considering the dynamics of confounding effects across samples (as shown in Equation 9), we naturally design a confounding score for each graph as $\omega(L) = rank(L)/N$ ($L$ is computed by equation 26. The graphs with high $\omega(L)$ can be regarded as confounding samples because the high rank of the matrix $L \in \mathbb{R}^{N \times D}$ stands for the extensive independent terms in $L$,

which indicates that sufficient exogenous confounding variables point to the $X$, and vice versa. Finally, the reconstruction error and ELBO can be encapsulated by:

$$l_{RC} = \omega(L)l_{rc}(X, \widehat{X} + C) + (1 - \omega(L))l_{rc}(X, \widehat{X})\mathcal{L} = l_{RC} - KL[q_\varphi(z|\mathcal{X})||p(z)] \tag{27}$$

Specifically, We adopt mean squared error (MSE) and in implementation:

$$\begin{aligned} l_{rc}(X, \widehat{X}) &= E_{q_\varphi(z|\mathcal{X})}[MSE(A, \hat{A})]l_{rc}(X, \widehat{X} + C) \\ &= E_{q_\varphi(z|\mathcal{X})}[MSE(X, (\hat{X} + C)] \end{aligned} \tag{28}$$

$$\tag{29}$$

## F.5 Details of Implementation

In ACD and AVICI, In our experiments, we used a Transformer encoder with a model size of 256 and a hidden size of 1024 for the feedforward modules. Throughout the training process, the learning rate was a constant base rate of $3 * 10^{-5}$. We optimized for a total of 300,000 primal steps, reducing the learning rate by a factor of ten after 200,000 steps. We adjusted the batch size based on the number of variables, ranging from 27 to 6.

In CAE and CVAE, throughout the training process, a learning rate of $3 * 10^{-5}$ was set. The batch size and epochs were set to 32 and 60, respectively. The implicit cause size was set to 192, the hidden size of the GNN was set to 300, and the dropout rate was 0.1. We set the number of layers (L) to 1. We evaluated our method ten times with different data splits on the test set.

In DAG-GNN, we used a variational autoencoder parameterized by a graph neural network. Throughout the training process, we used the Adam optimizer to solve the sub-problems of the augmented Lagrangian approach. When extracting the DAG, we used a threshold value of 0.45. The batch size and epochs were set to 32 and 80, respectively.

## Appendix G. Ablation Study about Disentanglement

To investigate whether the confounding effect is learned entirely by the $BL$ term and is independent of the causal effect, we conducted an ablation study to analyze the distance between the causal effect and the actual effect when the variable $L$ was intervened upon under different conditions. We continued to use the MSE between the predicted and actual causal representations as the metric for the causal representation. Our prediction target was the causal representation of target nodes with no children, and we analyzed the following ablation strategies:

1. "do front door": This strategy removes all paths where the confounding factor $L$ directly points to the target node. 2. "do back door": This strategy removes all outgoing paths from the confounding factor $L$ on the back-door path but connects the receiving node and the sending node of $L$ on this path. (This is equivalent to keeping the back-door path while removing the intermediary role of $L$.) 3. "do front and back": This strategy simultaneously removes both front-door and back-door paths. 4. "do pervasiveness": This strategy randomly reduces the number of edges between $L$ and the causal variable by half.

Table 14 presents the distance (MSE) between the causal representations predicted by these methods and the actual causal representations. The results clearly show that only the third ablation strategy ("do front and back"), which completely intervenes on the confounding effect, leads to the predicted causal effect being the closest to the actual causal effect. Strategy 1 and 2 could not fully eliminate the confounding effect, and thus their performance was not significantly different from the original

Table 14: Ablation study about disentanglement.

| Ablation | Confounder Number | | | |
|---|---|---|---|---|
| | 1 | 5 | 10 | 20 |
| original | 0.06 | 0.06 | 0.07 | 0.06 |
| do front door | 0.05 | 0.05 | 0.06 | 0.06 |
| do back door | 0.06 | 0.06 | 0.06 | 0.06 |
| do front and back | 0.01 | 0.01 | 0.01 | 0.02 |
| do pervasiveness | 0.09 | 0.05 | 0.08 | 0.08 |

nal data. In contrast, the fourth ablation strategy actually increased the confounding effect. This is because reducing the pervasiveness violates the sufficient statistic $C_j = E(X_j|L)$. This demonstrates that the $BL$ term can only be effectively estimated by our baseline when the confounding effect remains pervasive.

## Appendix H. Detailed Annotation results of Causaction

In this section, we present the annotation results for the causal relationships of each action in the Causaction dataset. The dataset records a total of ten behaviors: Coffee, Milk, Juice, Tea, Cereals, Fried Egg, Pancakes, Fruit Salad, Sandwich, and Scrambled Egg. Their specific action classifications are as follows:

**Coffee**: take cup → pour coffee → pour milk → pour sugar → spoon sugar → stir coffee

**Milk**: take cup → spoon powder → pour milk → stir milk

**Juice**: take squeezer → take glass → take plate → take knife → cut orange → squeeze orange → pour juice

**Tea**: take cup → add teabag → pour water → spoon sugar → pour sugar → stir tea

**Cereals**: take bowl → pour cereals → pour milk → stir cereals

**Fried Egg**: pour oil → butter pan → take egg → crack egg → fry egg → take plate → add salt and pepper → put egg onto plate

**Pancakes**: take bowl → crack egg → spoon flour → pour flour → pour milk → stir dough → pour oil → butter pan → pour dough into pan → fry pancake → take plate → put pancake onto plate

**Fruit Salad**: take plate → take knife → peel fruit → cut fruit → take bowl → put fruit into bowl → stir fruit

**Sandwich**: take plate → take knife → cut bun → take butter → smear butter → take topping → add topping → put bun together

**Scrambled Egg**: pour oil → butter pan → take bowl → crack egg → stir egg → pour egg into pan → stir fry egg → add salt and pepper → take plate → put egg onto plate

We show the detailed results of annotation in each behavior from Table 15 to Table 24. Additionally, the statistical annotation results show that the annotation probability of causal terms is rarely within the range of $[0.4, 0.6]$. Therefore, we can be confident that the annotation results are not significantly influenced by subjective factors arising from ambiguous causal relationships.

Table 15: The detailed results of Coffee behavior

| next action | take cup | pour coffee | pour milk | pour sugar | spoon sugar | stir coffee |
|---|---|---|---|---|---|---|
| take cup | 0 | 0 | 0 | 0 | 0 | 0 |
| pour coffee | 0.88 | 0 | 0 | 0 | 0 | 0 |
| pour milk | 0.79 | 0.09 | 0 | 0 | 0 | 0 |
| pour sugar | 0.21 | 0.22 | 0.15 | 0 | 0 | 0 |
| spoon sugar | 0 | 0.05 | 0.01 | 0.89 | 0 | 0 |
| stir coffee | 0.15 | 0.95 | 0.21 | 0.24 | 0.02 | 0 |

Table 16: The detailed results of Milk behavior

| next action | take cup | spoon powder | pour milk | stir milk |
|---|---|---|---|---|
| take cup | 0 | 0 | 0 | 0 |
| spoon powder | 0.05 | 0 | 0 | 0 |
| pour milk | 0.95 | 0.13 | 0 | 0 |
| stir milk | 0.14 | 0.11 | 0.96 | 0 |

Table 17: The detailed results of Juice behavior

| next action | take squeezer | take glass | take plate | take knife | cut orange | squeeze orange | pour juice |
|---|---|---|---|---|---|---|---|
| take squeezer | 0 | 0 | 0 | 0 | 0 | 0 | 0 |
| take glass | 0.09 | 0 | 0 | 0 | 0 | 0 | 0 |
| take plate | 0.09 | 0.11 | 0 | 0 | 0 | 0 | 0 |
| take knife | 0.05 | 0.05 | 0.055 | 0 | 0 | 0 | 0 |
| cut orange | 0 | 0.06 | 0.11 | 0.97 | 0 | 0 | 0 |
| squeeze orange | 0.87 | 0.05 | 0.01 | 0.08 | 0.26 | 0 | 0 |
| pour juice | 0.18 | 0.15 | 0.11 | 0.21 | 0.02 | 0.86 | 0 |

Table 18: The detailed results of Tea behavior

| next action | take cup | add teabag | pour water | spoon sugar | pour sugar | stir tea |
|---|---|---|---|---|---|---|
| take cup | 0 | 0 | 0 | 0 | 0 | 0 |
| add teabag | 0.67 | 0 | 0 | 0 | 0 | 0 |
| pour water | 0.95 | 0.87 | 0 | 0 | 0 | 0 |
| spoon sugar | 0.06 | 0.01 | 0.08 | 0 | 0 | 0 |
| pour sugar | 0.09 | 0.06 | 0.01 | 0.89 | 0 | 0 |
| stir tea | 0.24 | 0.99 | 0.85 | 0.04 | 0.08 | 0 |

Table 19: The detailed results of Cereals behavior

| next action | take bowl | pour cereals | pour milk | stir cereals |
|---|---|---|---|---|
| take bowl | 0 | 0 | 0 | 0 |
| pour cereals | 0.05 | 0 | 0 | 0 |
| pour milk | 0.94 | 0.03 | 0 | 0 |
| stir cereals | 0.04 | 0.91 | 0.06 | 0 |

Table 20: The detailed results of Fried Egg behavior

| next action | pour oil | butter pan | take egg | crack egg | fry egg | take plate | add salt and pepper | put egg onto plate |
|---|---|---|---|---|---|---|---|---|
| pour oil | 0 | 0 | 0 | 0 | 0 | 0 | 0 | 0 |
| butter pan | 0 | 0 | 0 | 0 | 0 | 0 | 0 | 0 |
| take egg | 0.05 | 0.05 | 0 | 0 | 0 | 0 | 0 | 0 |
| crack egg | 0.05 | 0.01 | 0.95 | 0 | 0 | 0 | 0 | 0 |
| fry egg | 0.06 | 0.01 | 0.75 | 0.79 | 0 | 0 | 0 | 0 |
| take plate | 0 | 0 | 0 | 0.05 | 0.13 | 0 | 0 | 0 |
| add salt and pepper | 0.07 | 0.15 | 0.08 | 0.05 | 0.86 | 0.12 | 0 | 0 |
| put egg onto plate | 0.05 | 0.05 | 0.67 | 0.77 | 0.12 | 0.02 | 0.02 | 0 |

Table 21: The detailed results of Pancakes behavior

| next action | take bowl | crack egg | spoon flour | pour flour | pour milk | stir dough | pour oil | butter pan | pour dough into pan | fry pancake | take plate | put pancake onto plate |
|---|---|---|---|---|---|---|---|---|---|---|---|---|
| take bowl | 0 | 0 | 0 | 0 | 0 | 0 | 0 | 0 | 0 | 0 | 0 | 0 |
| crack egg | 0.69 | 0 | 0 | 0 | 0 | 0 | 0 | 0 | 0 | 0 | 0 | 0 |
| spoon flour | 0.21 | 0.05 | 0 | 0 | 0 | 0 | 0 | 0 | 0 | 0 | 0 | 0 |
| pour flour | 0.68 | 0.19 | 0.89 | 0 | 0 | 0 | 0 | 0 | 0 | 0 | 0 | 0 |
| pour milk | 0.78 | 0.05 | 0.09 | 0.11 | 0 | 0 | 0 | 0 | 0 | 0 | 0 | 0 |
| stir dough | 0.01 | 0.05 | 0.05 | 0.67 | 0.11 | 0 | 0 | 0 | 0 | 0 | 0 | 0 |
| pour oil | 0.68 | 0.05 | 0.01 | 0 | 0.01 | 0.05 | 0 | 0 | 0 | 0 | 0 | 0 |
| butter pan | 0.1 | 0.04 | 0.03 | 0.07 | 0.04 | 0.95 | 0 | 0 | 0 | 0 | 0 | 0 |
| pour dough into pan | 0.05 | 0.11 | 0.12 | 0.06 | 0.98 | 0.87 | 0.95 | 0 | 0 | 0 | 0 | 0 |
| fry pancake | 0.11 | 0.05 | 0.05 | 0.31 | 0 | 0 | 0.77 | 0.75 | 0.99 | 0 | 0 | 0 |
| take plate | 0.12 | 0.05 | 0 | 0 | 0 | 0.15 | 0.01 | 0.02 | 0.05 | 0.08 | 0 | 0 |
| put pancake onto plate | 0 | 0 | 0 | 0 | 0 | 0.03 | 0.05 | 0.05 | 0.05 | 0.95 | 0.67 | 0 |

Table 22: The detailed results of Fruit Salad behavior

| next action | take plate | take knife | peel fruit | cut fruit | take bowl | put fruit into bowl | stir fruit |
|---|---|---|---|---|---|---|---|
| take plate | 0 | 0 | 0 | 0 | 0 | 0 | 0 |
| take knife | 0.02 | 0 | 0 | 0 | 0 | 0 | 0 |
| peel fruit | 0.02 | 0.91 | 0 | 0 | 0 | 0 | 0 |
| cut fruit | 0.05 | 0.85 | 0.95 | 0 | 0 | 0 | 0 |
| take bowl | 0 | 0.06 | 0.01 | 0.07 | 0 | 0 | 0 |
| put fruit into bowl | 0.07 | 0.06 | 0.71 | 0.88 | 0.76 | 0 | 0 |
| stir fruit | 0.05 | 0.02 | 0.11 | 0.01 | 0.02 | 0.96 | 0 |

Table 23: The detailed results of Sandwich behavior

| next action | take plate | take knife | cut bun | take butter | smear butter | take topping | add topping | put bun together |
|---|---|---|---|---|---|---|---|---|
| take plate | 0 | 0 | 0 | 0 | 0 | 0 | 0 | 0 |
| take knife | 0.01 | 0 | 0 | 0 | 0 | 0 | 0 | 0 |
| cut bun | 0.05 | 0.85 | 0 | 0 | 0 | 0 | 0 | 0 |
| take butter | 0.05 | 0.11 | 0.75 | 0 | 0 | 0 | 0 | 0 |
| smear butter | 0.16 | 0.11 | 0.85 | 0.79 | 0 | 0 | 0 | 0 |
| take topping | 0.05 | 0.06 | 0.04 | 0.05 | 0.13 | 0 | 0 | 0 |
| add topping | 0.07 | 0.15 | 0.08 | 0.05 | 0.06 | 0.92 | 0 | 0 |
| put bun together | 0.05 | 0.06 | 0.07 | 0.07 | 0.18 | 0.05 | 0.33 | 0 |

Table 24: The detailed results of Scrambled Egg behavior

| next action | pour oil | butter pan | take bowl | crack egg | stir egg | pour egg into pan | stir fry egg | add salt and pepper | take plate | put egg onto plate |
|---|---|---|---|---|---|---|---|---|---|---|
| pour oil | 0 | 0 | 0 | 0 | 0 | 0 | 0 | 0 | 0 | 0 |
| butter pan | 0.05 | 0 | 0 | 0 | 0 | 0 | 0 | 0 | 0 | 0 |
| take bowl | 0.01 | 0.03 | 0 | 0 | 0 | 0 | 0 | 0 | 0 | 0 |
| crack egg | 0.05 | 0.02 | 0.84 | 0 | 0 | 0 | 0 | 0 | 0 | 0 |
| stir egg | 0.07 | 0.05 | 0.33 | 0.94 | 0 | 0 | 0 | 0 | 0 | 0 |
| pour egg into pan | 0.05 | 0.26 | 0.12 | 0.88 | 0.95 | 0 | 0 | 0 | 0 | 0 |
| stir fry egg | 0.75 | 0.77 | 0.03 | 0.67 | 0.24 | 0.94 | 0 | 0 | 0 | 0 |
| add salt and pepper | 0.11 | 0.34 | 0.03 | 0.05 | 0.73 | 0.88 | 0.75 | 0 | 0 | 0 |
| take plate | 0.02 | 0.05 | 0.03 | 0.12 | 0.06 | 0.04 | 0.03 | 0.02 | 0 | 0 |
| put egg onto plate | 0.05 | 0.04 | 0.03 | 0.02 | 0.31 | 0.65 | 0.74 | 0.27 | 0.67 | 0 |

