# OpenReview forum: "Towards Causal Relationship in indefinite data: New Datasets and Baseline Model"
_DMLR — Accepted by DMLR_

### Review · Reviewer_QQcy · 2025-03-22

**Recommendation:** 3
**Confidence:** 3

**Summary Of Contributions:**

This paper introduces and addresses the concept of "indefinite data," characterized by multiple causal structures and complex (non-numerical) variables. The authors release two new datasets, Causalogue (text dialogue generated by GPT4) and Causaction (video actions), specifically designed for indefinite data. Additionally, they propose a probabilistic baseline model capable of learning both causal structures and representations simultaneously.

**Strengths:**

1. Clearly defined "indefinite data".
2. Contributed two annotated datasets for causal discovery.
3. Offered a probabilistic baseline model.

**Audience:**

Yes

**Claims And Evidence:**

Yes

**Datasets And Benchmarks:**

Yes

**Extended Submissions:**

No

**Limitations:**

1. The Causalogue dataset contains only four variables per sample, which is relatively small-scale. Moreover, it is generated using GPT-4, which may introduce biases or artifacts that affect the generalizability of the results.
2. The proposed probabilistic model relies heavily on assumptions, such as the linear ordering of variables.
3. The validation process for the representation of complex variables (Section 2.2) is unclear. Are the representations used to annotate unlabeled data or to validate existing annotations? If it is the latter, how are conflicts handled?
4. The causal formulations are not rigorous. For example, in Section 2.1, the input-output framework refers to a “causal model,” but it is not formally defined. Moreover, the formulation appears deterministic and omits noise terms.
5. The notion of causal representation in this paper is ambiguous. Is there a latent data-generating process behind it? The definition in Section 2.1 appears inconsistent with standard literature on causal representation learning.
6. It is unclear why the defined causal representation is expected to capture causal information rather than mere correlations.
7. The paper defines causal discovery as recovering the causal structure, but does not include the estimation of \\( \hat{X} \\) in Equation (1).
8. The identifiability of causal relationships is not guaranteed solely by acyclicity constraints. It is typically achieved through asymmetric patterns in the data—such as independence of noise terms. Furthermore, there are many identifiability results for cyclic models that are not considered here.

**Requested Changes:**

Please read established literature in causal discovery and causal representation learning and revise the formulations. See the above weaknesses for details.

**Strengths And Weaknesses:**

Strength:
1. Clearly defined "indefinite data".
2. Contributed two annotated datasets for causal discovery.
3. Offered a probabilistic baseline model.

Weaknesses:
1. The Causalogue dataset contains only four variables per sample, which is relatively small-scale. Moreover, it is generated using GPT-4, which may introduce biases or artifacts that affect the generalizability of the results.
2. The proposed probabilistic model relies heavily on assumptions, such as the linear ordering of variables.
3. The validation process for the representation of complex variables (Section 2.2) is unclear. Are the representations used to annotate unlabeled data or to validate existing annotations? If it is the latter, how are conflicts handled?
4. The causal formulations are not rigorous. For example, in Section 2.1, the input-output framework refers to a “causal model,” but it is not formally defined. Moreover, the formulation appears deterministic and omits noise terms.
5. The notion of causal representation in this paper is ambiguous. Is there a latent data-generating process behind it? The definition in Section 2.1 appears inconsistent with standard literature on causal representation learning.
6. It is unclear why the defined causal representation is expected to capture causal information rather than mere correlations.
7. The paper defines causal discovery as recovering the causal structure, but does not include the estimation of \\( \hat{X} \\) in Equation (1).
8. The identifiability of causal relationships is not guaranteed solely by acyclicity constraints. It is typically achieved through asymmetric patterns in the data—such as independence of noise terms. Furthermore, there are many identifiability results for cyclic models that are not considered here.

---

### Review · Reviewer_W8C2 · 2025-03-23

**Recommendation:** 3
**Confidence:** 2

**Summary Of Contributions:**

The paper presents a way to deal with indefinite data, i.e. data that are multi-structured and have complex variables.
The authors present two datasets to fill a gap in the literature regarding the lack of available indefinite data. They also introduce a probabilistic framework based on variational inference to deal with indefinite data. Several experiments have been carried out.

**Strengths:**

To the best of my knowledge, the work seems well-positioned in the literature. Also, the work is relevant to the research community, as it enhances the research on indefinite data.

**Audience:**

Yes

**Broader Impact Concerns:**

The authors discuss the broader impact.

**Claims And Evidence:**

The empirical evaluation supports the claims. However, I think the current version of the evaluation could greatly benefit from some polishing in the discussion and presentation.

**Datasets And Benchmarks:**

The datasheet is provided in the supplementary material.

**Extended Submissions:**

N/A

**Limitations:**

I listed in the requested changes a few current limitations of the paper.

**Requested Changes:**

I would ask the authors for the following changes:

1. regarding the experimental section, I think the paper would benefit from:
    - enlisting which research questions are being addressed with the empirical evaluation, and for each of such questions, discuss the results
    - I don't understand how the metrics (e.g., AUC, and MSE) are computed here. Could the authors provide some clarification?
    - Table 5 reports 95% intervals. How are these computed?

2. Discuss the notion of causality that is being used. I am not fully convinced by how the authors define it on page 10, i.e.,

    - > According to life experience, after action A happens, action B is high-probably to occur.
    - >According to life experience, after action B happens, action A is low probability to occur.

    I argue that such a notion considers only temporal precedence and does not address "what if" questions. Indeed, I argue that the causality notion should address a slightly different question: Would action B have occurred had action A not happened? Can the authors discuss this point?

3. I am also not convinced by the causal strength definition:
    - > Causal strength is defined as the value of a causal relationship. In this paper, we set the causal strength of each edge to be within [0, 1], also representing the probability of this edge being a causal relationship. For example, a causal strength of fi,j = 0.6 indicates a 60% probability of a causal relationship existing from node i to node j.

    I would argue that defining the existence of a causal relationship as a causal strength is misleading, as I would expect more of a causal effect quantification. Can the authors discuss this point?

4. Consider proper capitalization of letters in sections and paragraphs, including:
    - section 5, first paragraph;
    - section 6.2, second to last line missing reference
    - in the Appendix, several references/links to equations are missing, please adjust them

**Strengths And Weaknesses:**

The paper's main strengths are:

1. the approach is novel to the best of my knowledge;
2. the paper aims to fill some important gaps in terms of available data;
3. overall, I think the paper can be impactful in fostering current research on indefinite data

The paper's main weaknesses are:

1. clarity: the paper's clarity can be improved. For instance, there are several typos (e.g., words not properly capitalized, missing references, missing parentheses)
2. I am not sure about the notion of causality that is implemented in both datasets (more details in the limitations section)
3. the experimental section would benefit from a clearer structure

---

### Review · Reviewer_PFM8 · 2025-08-29

**Recommendation:** 3
**Confidence:** 2

**Summary Of Contributions:**

The paper introduces the notion of “indefinite data”, datasets that are simultaneously multi-structure (multiple DAGs across samples) and composed of complex variables (e.g., text or video, requiring learned representations). To bridge two gaps, it (i) releases two datasets—Causalogue (dialogue) and Causaction (video)—with pairwise causal annotations, and (ii) proposes a probabilistic VAE-style baseline that couples causal structure estimation and representation learning with latent confounders, formalized by the SCM 𝑋 = 𝐴𝑋 + 𝐵𝐿 + 𝐸 and a variational treatment of W=(I−A) ^−1 as the latent variable in the encoder–decoder (Eqs. 2–7). Experiments report advantages over amortized causal discovery and causal rep-learning baselines on structure recovery and “causal representation” quality, with OOD splits across structures/processes.

**Strengths:**

See Section Strengths and Weaknesses

**Audience:**

Yes

**Claims And Evidence:**

N/A

**Datasets And Benchmarks:**

N/A

**Extended Submissions:**

N/A

**Limitations:**

1. Formulation clarity. Treating causal saliency f_ij ∈[0,1] as the latent is a nice device for multi-structure coverage, but the prior over A or W=(I−A)^−1 and how sparsity/acyclicity are encouraged during variational inference could be specified more precisely (choice of prior, temperature/relaxation for edges, regularizers).
2. Decoder semantics. The decoder reconstructs X through f_2 (W(BL+E)). Because W is the Green’s function of I−A, the identifiability of A from W under noise and finite data merits discussion; multiple A can map to similar W if parameterized by continuous relaxations. A brief identifiability sketch would help.
3. Metrics for graphs. Reporting AUROC/MSE on adjacency matrices and Hamming distance is standard; consider also SHD and precision/recall at fixed thresholds, plus calibration of saliency estimates. The OOD split is appreciated; including per-structure breakdowns (e.g., which hybrids are hardest) would be informative.
4. Causal classifier f_c. The appendix shows f_c can learn directionality on simple-variable datasets, but those are not the target domain. Showing that f_c trained on Causalogue generalizes to held-out conversation styles or external corpora would increase confidence.
5. Acyclicity constraint. The NOTEARS-style trace constraint is proposed as an alternative. It would be useful to compare optimization stability and compute overhead between time-order and trace constraints, and to clarify the setting of α (spectral radius estimate) in practice.

**Requested Changes:**

1. For Causalogue, can you release the prompt templates and show that a model trained on your dataset does not merely learn template artifacts (e.g., via style transfer stress tests)? Any experiments on human-authored dialogues (e.g., DailyDialog) with causal re-annotation to assess transfer?
2. Beyond agreement rates, can you provide per-process confusion among annotator groups and consistency under perturbed video segments (e.g., dropped frames)? How robust is the “counterarrow” heuristic to class imbalance?
3. Have you tried jointly optimizing A and representations with a shared ELBO (rather than fixing A^)? Does it improve stability or accuracy, and how do you prevent degeneracy?
4. Under what assumptions on B, L, and E is disentanglement identifiable? Can you show interventional validation (e.g., simulate interventions on L and recover expected changes)?
5. Please tabulate every modification made to each baseline (architectures, losses, hyperparameters), and add sensitivity analyses showing your ranking is stable without those choices.
6. In the main results (not only Appendix B), could you add a variant without Assumption 2 as default, and report the delta relative to time-ordered training?
7. Can you test on independent indefinite datasets (e.g., causal graph reasoning on human dialogues or other action corpora) to measure true generalization beyond the two proposed datasets?

**Strengths And Weaknesses:**

Strengths: 1. The articulation of indefinite data clarifies why standard amortized CD methods (multi-structure but simple variables) and causal rep-learning (complex variables but single structure) fail when combined, i.e., the “model gap” (Sec. 4.1.3). This is a useful conceptual contribution for the community. 2. Causalogue: GPT-guided dialogues constructed from explicit causal patterns to reduce subjectivity in labels; labels are at the utterance-pair level. Causaction: video actions labeled causally at low-level segments with a multi-group quality-controlled process (200 qualified annotators, high agreement). These are the first datasets that are both multi-structure and complex-variable with comprehensive causal labels, per the paper’s survey. 3. The baseline cleanly merges structure learning and representation learning: estimate A first, then treat the estimate as “known” for representation learning, while explicitly modeling confounders L. Treating causal saliency (edge probabilities) as latent allows one model to cover multiple structures. This is technically neat and easy to extend. 4. Metrics on graph recovery (AUROC/MSE/Hamming) and representation causality via a learned causal classifier 𝑓_𝑐; OOD evaluation across unseen structures/processes (10-fold splits) is a strong test of generalization. The baseline wins consistently and with lower variance in OOD splits. 5. The paper discusses both a time-order assumption (Assumption 2) and a NOTEARS-style equality constraint as alternatives, reporting little loss in performance—useful for practical deployments where strict time order is unavailable.

Weaknesses: 1. Causalogue relies on GPT-generated dialogues following pre-specified patterns; this risks teaching to the test and may encode spurious regularities of the generator rather than natural conversational causality. More human-authored data or interventional probes would raise confidence. Causaction uses frequency-based heuristics (e.g., “counterarrow”) during annotation; while agreement is reported, ground truth causality in videos remains challenging, and the mapping from annotator frequency to causal edge truth deserves deeper validation. 2. The baseline is two-stage: estimate A, then representations using A^ as known. This raises endogeneity concerns, errors in A^ may bias representation learning; joint training with proper identifiability conditions or end-to-end amortization could be preferable. The paper acknowledges possible “causal inconsistency” but does not thoroughly analyze failure modes or provide guarantees. 3. Causal representation quality is assessed through a learned classifier f_c trained on the same datasets (albeit with splits). Without an external ground truth of causal factors for complex variables, this risks circularity; ablations showing invariance to context distribution shifts or performance on downstream interventional tasks would be more convincing. 4. Several competing methods are modified (e.g., hidden dimension increases, loss mapped into “correlation relationship space,” latent variables replaced by causal saliency). Although arguably necessary, these adaptations may disadvantage baselines—or at least make comparisons less apples-to-apples—so the fairness of comparisons is not airtight. Detailed sensitivity analyses are warranted. 5. The time-order assumption (Assumption 2) strongly constrains admissible graphs; many real indefinite datasets (e.g., multi-speaker dialogues with interruptions, or cyclical physical processes) violate linear orders. The appendix addresses an acyclicity constraint, but most results lean on time order. More results without Assumption 2 as default would strengthen claims. 6. While training details for encoders (RoBERTa/VideoMAE) and splits are given, the exact templates/patterns used to generate Causalogue, inter-annotator matrix for Causaction per process, and hyperparameters for all modified baselines are not fully enumerated in the main text. This limits reproducibility and external auditing of labeling pipelines and baselines. 7. The paper claims disentanglement of confounding via BL, but quantitative evidence appears limited to a synthetic dataset summary; stronger demonstrations (e.g., interventions on L, or controlled confounding strength sweeps) would substantiate this key claim.